

# Efficiency and robustness in Monte Carlo sampling of 3-D geophysical inversions with Obsidian v0.1.2: Setting up for success

Richard Scalzo[1], David Kohn[2], Hugo Olierook[3], Gregory Houseman[4], Rohitash Chandra[1,5], Mark Girolami[6,7], and Sally Cripps[1,8]

[1]Centre for Translational Data Science, University of Sydney, Darlington NSW 2008, Australia
[2]Sydney Informatics Hub, University of Sydney, Darlington NSW 2008, Australia
[3]School of Earth and Planetary Sciences, Curtin University, Bentley WA 6102, Australia
[4]School of Earth and Environment, University of Leeds, Leeds, LS2 9JT, UK
[5]School of Geosciences, University of Sydney, Darlington NSW 2008, Australia
[6]The Alan Turing Institute for Data Science, British Library, 96 Euston Road, London, NW1 2DB, UK
[7]Department of Mathematics, Imperial College London, London, SW7 2AZ, UK
[8]School of Mathematics and Statistics, University of Sydney, Darlington NSW 2008, Australia

**Correspondence:** Richard Scalzo (richard.scalzo@sydney.edu.au)

**Abstract.** The rigorous quantification of uncertainty in geophysical inversions is a challenging problem. Inversions are often ill-posed and the likelihood surface may be multimodal; properties of any single mode become inadequate uncertainty measures, and sampling methods become inefficient for irregular posteriors or high-dimensional parameter spaces. We explore the influences of different choices made by the practitioner on the efficiency and accuracy of Bayesian geophysical inversion

methods that rely on Markov chain Monte Carlo sampling to assess uncertainty, using a multi-sensor inversion of the three-dimensional structure and composition of a region in the Cooper Basin of South Australia as a case study. The inversion is performed using an updated version of the Obsidian distributed inversion software. We find that the posterior for this inversion has complex local covariance structure, hindering the efficiency of adaptive sampling methods that adjust the proposal based on the chain history. Within the context of a parallel-tempered Markov chain Monte Carlo scheme for exploring high-dimensional

multi-modal posteriors, a preconditioned Crank-Nicholson proposal outperforms more conventional forms of random walk. Aspects of the problem setup, such as priors on petrophysics or on 3-D geological structure, affect the shape and separation of posterior modes, influencing sampling performance as well as the inversion results. Use of uninformative priors on sensor noise can improve inversion results by enabling optimal weighting among multiple sensors even if noise levels are uncertain. Efficiency could be further increased by using posterior gradient information within proposals, which Obsidian does not currently

support, but which could be emulated using posterior surrogates.



## 1  Introduction

Construction of 3-D geological models is plagued by the limitations on direct sampling and geophysical measurement (Wellmann et al., 2010; Lindsay et al., 2013). Direct geological observations are sparse because of the difficulty in acquiring them, being often obscured by sedimentary or regolith cover; resolving this issue via drilling is expensive (Anand and Butt, 2010;

Salama et al., 2016). Indirect observations via geophysical sensors deployed at or above the surface are more readily obtained (Strangway et al., 1973; Gupta and Grant, 1985; Sabins, 1999; Nabighian et al., 2005b, a). However, gravity, magnetic, and electrical measurements integrate data from the surrounding volume, so it is difficult to resolve precise geological constraints at any given position and depth, except where borehole measurements are also available. Determining the true underlying geological structure, or range of geological structures, consistent with observations constitutes an often poorly constrained

inverse problem. One natural way to approach this is forward-modelling, where the responses of various sensors on a proposed geological structure are simulated, and the proposed structure is then updated or sampled iteratively (for examples see Jessell, 2001; Calcagno et al., 2008; Olierook et al., 2015).

  The incompleteness and uncertainty of the information contained in any single data set frequently mean that there are many possible worlds consistent with the data being analyzed (Tarantola and Valette, 1982; Tarantola, 2005). To the extent that in-

formation provided by different datasets is complementary, combining all available information into a single joint inversion reduces uncertainty in the final results. Accomplishing this in a principled and self-consistent manner presents several challenges, including: (i) how to weigh constraints provided by different datasets relative to each other; (ii) how to rule out worlds inconsistent with geological processes (expert knowledge); (iii) how to present a transparent accounting of the remaining uncertainty; and (iv) how to do all this in a computationally efficient manner.

Bayesian statistical techniques provide a powerful framework for characterizing and fusing disparate sources of probabilistic information (Tarantola and Valette, 1982; Mosegaard and Tarantola, 1995; Sambridge and Mosegaard, 2002; Sambridge et al., 2012). All input sources of information — from geophysical sensors, geological field observations, previous inferences, or expert knowledge — are treated as probability distributions; this forces the practitioner to make explicit all assumptions, not only about expected values, but about uncertainties. The output of a Bayesian method is also a probability distribution (the

*posterior*), for which the gold-standard representation is a set of samples from a Monte Carlo algorithm, in particular *Markov chain Monte Carlo* (MCMC; Mosegaard and Tarantola, 1995; Sambridge and Mosegaard, 2002). The posterior distribution is a representation of all possible outcomes and hence provides an internal estimate of uncertainty. The uncertainty associated with the posterior can be visualized in terms of the marginal distributions of parameters of interest, or rendered in 3-D voxelisations of information entropy (Wellmann and Regenauer-Lieb, 2012). The posterior also can be readily updated online as new

information becomes available, making Bayesian approaches optimal for decision-making under risk and uncertainty.

  Although Bayesian methods provide rigorous uncertainty quantification, implementing them in practice for complicated forward models with many free parameters has proven difficult in other geoscientific contexts, such as landscape evolution (Chandra et al., 2018) and coral reef assembly (Pall et al., 2018). Sambridge and Mosegaard (2002) point out the challenge of capturing all elements of a geophysical problem in terms of probability, which can be difficult for complex datasets and



even harder for approximate forward models or world representations where the precise nature of the approximation is hard to capture. The irregular shapes and multimodal structure of the posterior distributions for realistic geophysics problems makes them hard to explore; the second moment (variance) of the posterior around each local maximum may in these cases significantly underestimate uncertainties. Moreover, the large number of parameters needed to specify 3-D structures also means

these irregular posteriors are embedded in high-dimensional spaces, increasing the computational cost for both optimization and sampling. Therefore, the sampling methods must usually be tailored to each individual problem and no "one-size-fits-all" solution exists.

These limitations form the backdrop for current work on applying Bayesian principles to 3-D structural modelling. Giraud et al. (2017, 2018) demonstrate an optimization-based Bayesian inversion framework for 3-D geological models, which finds

the maximum of the posterior distribution (*maximum a posteriori*, or MAP), and expresses uncertainty in terms of the posterior covariance around the MAP solution; while they show that fusing data reduces uncertainty around this mode, they do not attempt to find or characterize other modes, or higher moments of the posterior. Ruggeri et al. (2015) investigate several MCMC schemes for sampling a single-sensor inverse problem (crosshole georadar travel time tomography), focusing on sequential, localized perturbations of a proposed 3-D model ("sequential geostatistical resampling", or SGR); they show that sampling

is impractically slow due to high dimensionality and correlations between model parameters. Laloy et al. (2016) embed the SGR proposal within a parallel-tempered sampling scheme to explore multiple posterior modes of a 2-D inverse problem in groundwater flow, improving computational performance but not to a cost-effective threshold. The above methods are non-parametric, in that the model parameters simply form a 3-D field of rock properties to which sensors respond. de la Varga and Wellmann (2016); de la Varga et al. (2018) focus on building parametrized 3-D models in order to reduce the problem

dimension and to naturally incorporate structural measurements, but have not yet tested this framework on a large-scale 3-D joint inversion with multiple sensors.

McCalman et al. (2014a) present Obsidian, a flexible software platform for MCMC sampling of 3-D multi-modal geophysical models on distributed computing clusters. Beardsmore et al. (2016) demonstrate Obsidian on a test problem in geothermal exploration, in the Moomba gas field of the Cooper Basin in South Australia, comparing their results to a deterministic inversion

of the same area performed by Meixner and Holgate (2009). These papers outline a full-featured open-source inversion method that can fuse heterogeneous data into a detailed solution, but make few comments about how the efficiency and robustness of the method depends on the particular choices they made.

In this paper, we revisit the inversion problem of Beardsmore et al. (2016) using a customized version (Scalzo et al., 2019) of the McCalman et al. (2014a) inversion code. Our interest is in exploring this problem as a case study to determine which

aspects of this problem's posterior present the most significant obstacles to efficient sampling, which updates to the MCMC scheme improve sampling under these conditions, and how plausible alternative choices of problem setup might influence the efficiency of sampling or the robustness of the inversion. The aspects we consider include: correlations between model parameters; relative weights between datasets with poorly constrained uncertainty; and choices of priors representing different possible exploration scenarios.





## 2 Background

In this section we present a brief overview of the Bayesian forward-modeling paradigm to geophysical inversions. We also provide a discussion of implementing Bayesian inference via sampling using MCMC methods. We then present background on the original Moomba inversion problem, commenting on choices made in the inversion process before we begin to explore
different choices in subsequent sections.

### 2.1 Overview of Bayesian inversion

A Bayesian inversion scheme for geophysical forward models comprises of three key elements:

1. the underlying parametrized representation of the simulated volume or history, which we call the *world* or *world view*, denoted by a vector of *world parameters* $\boldsymbol{\theta} = (\theta_1, \ldots, \theta_P)$

2. a probability distribution $p(\boldsymbol{\theta})$ over the world parameters, called the *prior*, expressing expert knowledge or belief about the world before any datasets are analyzed; and

3. a probability distribution $p(\mathcal{D}|\boldsymbol{\theta})$ over possible realizations of the observed data $\mathcal{D}$ as a function of world parameters, called the *likelihood*, that incorporates the prediction of a deterministic forward model $g(\boldsymbol{\theta})$ of the sensing process for each value of $\boldsymbol{\theta}$.

The *posterior* is then the distribution $p(\boldsymbol{\theta}|\mathcal{D})$ of values of the world parameters consistent with both prior knowledge and observed data. *Bayes' theorem* describes the relationship between the prior, likelihood, and posterior:

$$p(\boldsymbol{\theta}|\mathcal{D}) = \frac{p(\mathcal{D}|\boldsymbol{\theta})p(\boldsymbol{\theta})}{\int p(\mathcal{D}|\boldsymbol{\theta})p(\boldsymbol{\theta}) \, d\boldsymbol{\theta}}. \tag{1}$$

Our terminology for these elements is typical of the statistics literature, so it is critical to identify the same elements in terminology used in previous geophysical inversion papers (for example Menke, 2018). In previous papers a "model" might
refer to the world representation, whereas below we will use the word "model" to refer to the *statistical* model defined by a choice of all of the above elements. A non-Bayesian inversion would proceed by minimizing an *objective function*, one simple form of which is the mean square misfit between the (statistical) model predictions and the data, corresponding to our negative log likelihood (for observational errors that are independent and Gaussian-distributed with precisely known variance). To penalize solutions that are considered *a priori* unlikely, the objective function might include additional *regularization* terms
corresponding to the negative log priors in our framework. The full objective function would thus correspond to our negative log posterior, and minimization of the objective function would correspond to maximization of our posterior probability, under some choice of prior. However, regularization does not necessarily proceed from a probabilistic interpretation; interpreting model elements in terms of probability may motivate different choices of likelihood or prior than the usual non-probabilistic misfit or regularization terms.

Indeed, there is considerable flexibility in choosing the above elements even in a fully probabilistic context. For example, the partitioning of information into "data" and "prior knowledge" is neither unique nor cut-and-dried. However, there are





guiding principles: the ideal set of parameters $\boldsymbol{\theta}$ is both *parsimonious* — as few as possible to faithfully represent the world — and *interpretable*, referring to meaningful aspects of the world that can easily be read off the parameter vector. Information resulting from processes that can be easily simulated belong in the likelihood: for example, one might argue that the output of a gravimeter should have a Gaussian distribution, because it responds to the mean rock density within a volume and hence obeys

the central limit theorem, or that the output of a Geiger counter should follow a Poisson distribution to reflect the physics of radioactive decay. Even processes that are not so easily simulated can at least be approximately described, for example by using a mixture distribution to account for outlier measurements (Mosegaard and Tarantola, 1995) or a prior on the unknown noise level in a process Sambridge et al. (2012). Other information about allowable or likely worlds belongs in the prior, such as the distribution of initial conditions for simulation, or interpretations of datasets with expensive or intractable forward models.

The inference process expresses its results in terms either of $p(\boldsymbol{\theta}|\mathcal{D})$ itself or of integrals over $p(\boldsymbol{\theta}|\mathcal{D})$ (including credible limits on $\boldsymbol{\theta}$). This is different from the use of point estimates for the world parameters, such as the *maximum likelihood* (ML) solution $\boldsymbol{\theta}_{\mathrm{ML}} = \sup_{\boldsymbol{\theta}} p(\mathcal{D}|\boldsymbol{\theta})$ or the *maximum a posteriori* (MAP) solution $\boldsymbol{\theta}_{\mathrm{MAP}} = \sup_{\boldsymbol{\theta}} p(\mathcal{D}|\boldsymbol{\theta})p(\boldsymbol{\theta})$. To the extent that ML or MAP prescriptions give any estimate of uncertainty on $\boldsymbol{\theta}$, they usually do so through the covariance of the log likelihood or log posterior around the optimal value of $\boldsymbol{\theta}$, equivalent to a local approximation of the likelihood or posterior by a multivariate

Gaussian. As mentioned above, these approaches will underestimate the uncertainty for complex posteriors; a more rigorous accounting of uncertainty will include all known modes, higher moments of the distribution, or (more simply) providing enough samples from the distribution to characterize it.

The posterior distribution $p(\boldsymbol{\theta}|\mathcal{D})$ is rarely available in closed form. However, it is often known up to a normalizing constant: $p(\mathcal{D}|\boldsymbol{\theta})p(\boldsymbol{\theta})$. Sampling methods such as MCMC can therefore be used to approximate the posterior, without having to explicitly

evaluate the normalizing constant (the high-dimensional integral in the denominator of Eq. 1). It is to these methods we turn next.

## 2.2 Markov chain Monte Carlo

A MCMC algorithm comprises a sequence of world parameter vectors $\{\boldsymbol{\theta}^{[j]}\}$, called a *(Markov) chain*, and a *proposal distribution* $q(\boldsymbol{\theta}'|\boldsymbol{\theta})$ to generate a new set of parameters based only on the last element of the chain. In the commonly-used

*Metropolis-Hastings algorithm* (Metropolis et al., 1953; Hastings, 1970), a proposal $\boldsymbol{\theta}' \sim q(\boldsymbol{\theta}'|\boldsymbol{\theta}^{[j]})$ is at random either *accepted* and added to the chain's history ($\boldsymbol{\theta}^{[j+1]} = \boldsymbol{\theta}'$) with probability

$$P_{\mathrm{accept}} = \min\left(1, \frac{P(\mathcal{D}|\boldsymbol{\theta}')P(\boldsymbol{\theta}')q(\boldsymbol{\theta}^{[j]}|\boldsymbol{\theta}')}{P(\mathcal{D}|\boldsymbol{\theta}^{[j]})P(\boldsymbol{\theta}^{[j]})q(\boldsymbol{\theta}'|\boldsymbol{\theta}^{[j]})}\right), \tag{2}$$

or *rejected* and a copy of the previous state added instead ($\boldsymbol{\theta}^{[j+1]} = \boldsymbol{\theta}^{[j]}$). This rule guarantees, under certain regularity conditions (Chib and Greenberg, 1995), that the sequence $\{\boldsymbol{\theta}^{[j]}\}$ converges to the required stationary distribution, $P(\boldsymbol{\theta}|\mathcal{D})$, in the

limit of increasing $n$.

Metropolis-Hastings algorithms form a large class of sampling algorithms, limited only by the forms of proposals. Although proofs that the chain will *eventually* sample from the posterior are important, clearly chains based on *efficient* proposals are to





be preferred. A proposal's efficiency will depend on the degree of correlation between consecutive states in the chain, which in turn can depend on how well matched the proposal distribution is to the properties of the posterior.

One simple, commonly used proposal distribution is a (multivariate) *Gaussian random walk* (GRW) step $u$ from the chain's current position, drawn from a multivariate Gaussian distribution with covariance matrix $\boldsymbol{\Sigma}$:

$$\boldsymbol{\theta}' = \boldsymbol{\theta}^{[j]} + \boldsymbol{u}, \qquad \boldsymbol{u} \sim N(\mathbf{0}, \boldsymbol{\Sigma}). \tag{3}$$

This proposal is straightforward to implement, but its effectiveness can depend strongly on $\boldsymbol{\Sigma}$, and does not in general scale well to rich, high-dimensional world parametrizations. If $\boldsymbol{\Sigma}$ has too large a scale, the GRW proposal will step too often into regions of low probability, resulting in many repeated states due to rejections; if the scale is too small, the chain will take only small, incremental steps. In both cases, subsequent states are highly correlated. If the shape of $\boldsymbol{\Sigma}$ is not tuned to capture correlations between different dimensions of $\boldsymbol{\theta}$, the overall scale must usually be reduced to ensure a reasonable acceptance fraction.

The SGR method (Ruggeri et al., 2015; Laloy et al., 2016) can be seen as a mixture of multivariate Gaussians, in which $\boldsymbol{\Sigma}$ has highly correlated sub-blocks of parameters, corresponding to variations of the world over different spatial scales. Ruggeri et al. (2015) and Laloy et al. (2016) evaluate SGR using single-sensor inversions in crosshole georadar travel time tomography, with posteriors corresponding to a Gaussian process — an unusually tractable (if high-dimensional) problem that could be solved in closed form as a cross-check. These authors found that in general updating blocks of parameters simultaneously was inefficient, which may not be surprising in a high-dimensional model: for a tightly constrained posterior lying along a low-dimensional subspace of parameter space, almost all directions — hence almost all posterior covariance choices — lead towards regions of low probability. Directions picked at random without regard for the shape of the posterior will scale badly with increasing dimension.

Many other types of proposals exist, using information from ensembles of particles (Goodman and Weare, 2010), adaptation of the proposal distribution based on the chain's history (Haario et al., 2001), derivatives of the posterior (Neal et al., 2011; Girolami and Calderhead, 2011), approximations to the posterior (Strathmann et al., 2015), and so forth. The GRW proposal is not only easy to write down and fast to evaluate, but requires no derivative information. We will compare and contrast several derivative-free proposals in our experiments below.

The posterior distributions arising in geophysical inversion problems are also frequently multi-modal; MCMC algorithms to sample such posteriors need the ability to escape from, or travel easily between, local modes. *Parallel-tempered MCMC* PTMCMC (Geyer and Thompson, 1995) is a meta-method for sampling multi-modal distributions that works by running an ensemble of Markov chains. The ensemble is characterized by a sequence of $M+1$ parameters $\{\beta_i\}$, with $\beta_0 = 1 > \beta_1 > \beta_2 > \ldots > \beta_M > 0$, called the *(inverse) temperature ladder*. Each chain samples the distribution

$$P_i(\boldsymbol{\theta}|D) \propto (P(D|\boldsymbol{\theta}))^{\beta_i} P(\boldsymbol{\theta}), \tag{4}$$

so that the chain with $\beta_0 = 1$ is sampling from the desired posterior, and a chain with $\beta_i = 0$ samples from the prior, which should be easy to explore. Chains with intermediate values $0 < \beta < 1$ sample intermediate distributions in which the data's





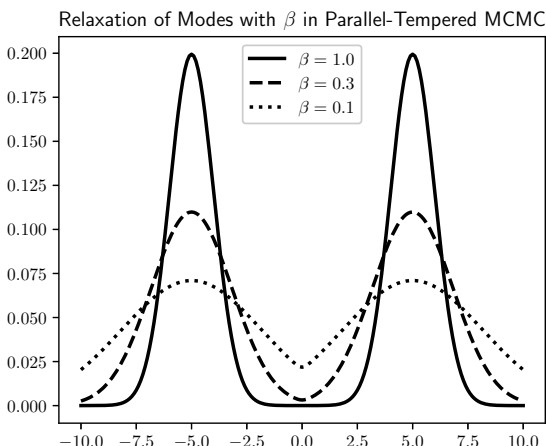

**Figure 1.** Parallel-tempered relaxation of a bimodal distribution.

influence is reduced, so that modes are shallower and easier for chains to escape and traverse. In addition to proposing new states within each chain, PTMCMC includes Metropolis-style proposals that allow adjacent chains on the temperature ladder, with inverse temperatures $\beta$ and $\beta'$, to swap their most recent states $\boldsymbol{\theta}$ and $\boldsymbol{\theta}'$ with probability

$$P_{\mathrm{swap}} = \min\left(1, \left[\frac{P(\mathcal{D}|\boldsymbol{\theta}')}{P(\mathcal{D}|\boldsymbol{\theta})}\right]^{\beta'-\beta} \frac{P(\boldsymbol{\theta}')}{P(\boldsymbol{\theta})}\right). \tag{5}$$

This allows chains with current states spread throughout parameter space to share global information about the posterior in such a way that chain $i$ still samples $P_i(\boldsymbol{\theta}|D)$ in the long-term limit. The locations of discovered modes diffuse from low-$\beta_i$ chains (which can jump freely between relaxed, broadened versions of these modes) towards the $\beta_0 = 1$ chain, which can then sample from all modes of the unmodified posterior in the correct proportions. The temperature ladder should be defined so that adjacent chains on the ladder are sampling from distributions similar enough for swaps to occur frequently.

Figure 1 illustrates the sampling of a simple bimodal probability distribution (a mixture of two Gaussians) via PTMCMC. The solid line depicts the true bimodal distribution, while the broken lines shows the stationary distribution of tempered chains for smaller values of $\beta$. The tempered chains are more likely to propose moves across modes than the untempered chains, and the existence of a sequence of chains ensures that the difference in probability between successive chains is small enough that swaps can take place easily.

Even without regard to multiple modes, PTMCMC can also help to reduce correlations between successive independent posterior samples. Laloy et al. (2016) use SGR as a within-chain proposal in a PTMCMC scheme, demonstrating its effects on correlations between samples but noting that the algorithm remains computationally intensive.





## 2.3 Performance metrics for MCMC

Because MCMC guarantees results only in the limit of large samples, criteria are still required to assess the algorithm's performance. Suppose for the discussion below that up to the assessment point, we have obtained $N$ samples of a $d$-dimensional posterior from each of $M$ separate chains; let $\boldsymbol{\theta}_i^{[j]} = (\theta_{1i}^{[j]}, \ldots, \theta_{di}^{[j]})$ be the $d \times 1$ vector of parameter values drawn at iteration $[j]$ in chain $i$. Let

$$\hat{\theta}_{ki} = \frac{1}{N} \sum_{j=1}^{N} \theta_{ki}^{[j]}$$

be the mean value of the parameter $\theta_k$ in chain $i$, across the $N$ iterates, and let $\tilde{\theta}_k = \frac{1}{M} \sum_{i=1}^{M} \hat{\theta}_{ki}$ be the sample mean of $\theta_k$ across all iterates and chains. Then

$$B_k = \frac{1}{M-1} \sum_{i=1}^{M} (\hat{\theta}_{ki} - \tilde{\theta}_k)^2$$

Further define

$$s_{ki}^2 = \frac{1}{M-1} \sum_{j=1}^{N} (\theta_{ki}^{[j]} - \hat{\theta}_{ki})^2$$

and

$$W_k = \frac{1}{M} \sum_{i=1}^{M} s_{ki}^2.$$

For Metropolis-Hastings MCMC, the *acceptance fraction* of proposals is easily measured, and for a chain that is performing well should be $\sim 20$–$50\%$. Roberts et al. (1997) showed that the optimal acceptance fraction for random walks in the limit of a large number of dimensions is 0.234, which we will take as our target since the proposals we will consider are modified random walks.

We examine correlations between samples within each chain separated by a lag time $l$ using the *autocorrelation function*,

$$\rho_{lki} = \frac{1}{(N-l)W_k} \sum_{j=l+1}^{N} (\theta_{ki}^{[j]} - \hat{\theta}_{ki})(\theta_{ki}^{[j-l]} - \hat{\theta}_{ki}), \tag{6}$$

The number of independent draws from the posterior with equal statistical power to each set of $N$ chain samples scales with the area under the autocorrelation function or *(integrated) autocorrelation time* (IACT),

$$\tau_{ki} = 1 + 2 \sum_{l=1}^{N} \left(1 - \frac{k}{N}\right) \rho_{lki}. \tag{7}$$

A *trace plot* of the history of an element of the parameter vector $\boldsymbol{\theta}$ over time summarizes the sampling performance at a glance, revealing where in parameter space an algorithm is spending its time; Fig. 2 shows a series of such figures for some of the different MCMC runs in the present work.



Gelman and Rubin (1992) assess the number of samples required to reach a robust sampling of the posterior by comparing results among multiple chains. If the simulation has run long enough, the mean values among chains should differ by some small fraction of the width of the distribution; intuitively, this is similar to a hypothesis test that the chains are sampling the same marginal distribution for each parameter. More precisely, the quantity

$$\hat{V}_k/W_k = \frac{N-1}{N} + \frac{M+1}{MN}B_k/W_k \tag{8}$$

provides a metric for convergence of different chains to the same result, which decreases to 1 as $N \to \infty$. The chains may be stopped and results read out when the metric dips below a target value for all world parameters $\boldsymbol{\theta}$. The precise number of samples needed may depend on the details of the distribution; the metric provides a stopping condition, but not an estimate of how long it will take to achieve.

The results from this procedure must still be evaluated according to how well the underlying statistical model describes the geophysical data, and whether the results are geologically plausible — although this is not unique to MCMC solutions. The distribution of residuals of model predictions (forward-modeled data sets) from the observed data can be compared to the assumed likelihood. The standard deviation or variance of the residuals (relative to the uncertainty) provide a convenient single-number summary, but the spatial distribution of residuals may also be important; outliers and/or structured residuals will indicate places where the model fails to predict the data well, and highlight parts of the model parametrization that need refinement.

Finally, representative instances of the world itself should be visualized to check for surprising features. Given the complexity of real-world data, the adequacy of a given model is in part a matter of scientific judgment, or fitness for a particular applied purpose to which the model will be put. We will use the term *model inadequacy* to refer to model errors arising from approximations or inaccuracies in the world parametrization or the mathematical specification of the forward model — although there will always be such approximations in real problems, and the presence of model inadequacy should not imply that the model is unfit for purpose.

### 2.4 The Obsidian distributed PTMCMC code

For our experiments we use a customized fork (v0.1.2 Scalzo et al., 2019) of the open-source Obsidian software package. Obsidian was previously presented in McCalman et al. (2014a) and was used to obtain the modeling results of Beardsmore et al. (2016); v0.1.1 was the most recent open-source version publicly available before our work. We refer the reader to previous publications for a comprehensive description of Obsidian, but below we summarize key elements corresponding to the inversion framework set out above.

**World parametrization:** Obsidian's world is parametrized as a series of discrete layers, each with its own spatially constant rock properties, separated by smooth boundaries. Each layer boundary is a two-dimensional Gaussian process regression against a set of *control points* that specify the subsurface depth of the boundary at given surface locations. The layer boundaries are indexed in order of increasing depth beneath the surface, but are allowed to cross over each other. In regions where the formal layer thickness $z_{i+1} - z_i$ is negative, the corresponding rock layer pinches out to zero thickness. For a world with $N$ layers,





indexed by $i$ with $1 \leq i \leq N$, each with a grid $n_i$ regularly spaced control points at sites $x_i$ and rock properties corresponding to each of $S$ forward-modeled sensors, the parameter vector is therefore

$$\boldsymbol{\theta} = (\alpha_{11} \dots \alpha_{Nn_N}, \rho_{11} \dots \rho_{NK}), \tag{9}$$

where $\alpha_{ij}$ is the offset of the mean depth of the top of layer $i$ at site $j$, and $\rho_{is}$ is the rock property of layer $i$ associated with

sensor $s$. Taken together, the rock properties for each layer and the control points for the boundaries between the layers fully specify the world. This parametrization requires that interface depths be single-valued, not for example permitting the surface to fold above or below. Such a limitation still enables reasonable representations of sedimentary basins, but may hinder faithful modeling of other kinds of structures.

**Prior:** The control point depth offsets within each layer $i$ have a multivariate Gaussian prior with mean zero and covariance

$\boldsymbol{\Sigma_{\alpha_i}}$. The Gaussian processes which interpolate the layer boundaries across the lateral extent of the world use a radial basis function kernel,

$$k(x, y; x', y') = \exp\left(-\frac{(x-x')^2}{\Delta_x^2} - \frac{(y-y')^2}{\Delta_y^2}\right), \tag{10}$$

and has mean function $\mu_i(x, y)$ that can be specified at finer resolution to capture fine detail in layer structure. The correlation lengths $\Delta_x$ and $\Delta_y$ could in principle be varied, but in this case are fixed in value to the spacing between control point locations

along the $x$ and $y$ coordinate axes, respectively. The rock properties for each layer $i$, which are statistically independent of the control points, also have a multivariate Gaussian prior, with mean $\boldsymbol{\mu_{\rho_i}}$ and covariance $\boldsymbol{\Sigma_{\rho_i}}$). The prior for the full parameter vector is therefore block-diagonal,

$$
\begin{aligned}
P(\boldsymbol{\theta}) &= \prod_{i=1}^{N} P(\boldsymbol{\alpha_{i\cdot}}) P(\boldsymbol{\rho_{i\cdot}}) \\
&= \prod_{i=1}^{N} N(\boldsymbol{\alpha_{i\cdot}}; 0, \boldsymbol{\Sigma_{\alpha_i}}) N(\boldsymbol{\rho_{i\cdot}}; \boldsymbol{\mu_{\rho_i}}, \boldsymbol{\Sigma_{\rho_i}}).
\end{aligned} \tag{11}
$$

**Likelihood:** The likelihood for each Obsidian sensor $s$ is Gaussian, meaning that the residuals of the data $\mathcal{D}_s$ from the forward model predictions $f_s(\boldsymbol{\theta})$ for the true world parameters $\theta$ are assumed to be independent, identically distributed Gaussian draws. The underlying variance of the Gaussian noise is not known, but is assumed to follow an inverse gamma distribution $\mathrm{IG}(x; \alpha_s, \beta_s)$ with different (user-specified) hyperparameters $\alpha_s, \beta_s$ for each sensor $s$. This choice of distribution amounts to a prior, but the hyperparameters $\alpha_s$ and $\beta_s$ for each sensor are not explicitly sampled over; instead, they are integrated out

analytically, so that the final likelihood has the form

$$P(\mathcal{D}_s | \boldsymbol{\theta}) = \prod_{k=1}^{K_s} t_{2\alpha_s}\left(\frac{\beta_s}{\alpha_s}(f_s(\boldsymbol{\theta}) - \mathcal{D}_s)\right), \tag{12}$$

where $t_\nu(x)$ is a Student's-$t$ distribution with $\nu$ degrees of freedom. This distribution is straightforward to calculate, although the results may be sensitive to the user's choices of $\alpha_s$ and $\beta_s$; unrestrictive choices (e.g. $\alpha_s = \beta_s = 1$) should be used if the



user has little prior knowledge about the noise level in the data. The likelihood including all sensors is therefore

$$P(\mathcal{D}|\boldsymbol{\theta}) = \prod_{s=1}^{S} P(\mathcal{D}_s|\boldsymbol{\theta}), \tag{13}$$

since each sensor probes a different physical aspect of the rock.

**MCMC:** The sampling algorithm used by Obsidian is an adaptive form of PTMCMC, described in detail in Miasojedow
et al. (2013). This algorithm allows for the progressive adjustment of the step size used for proposals within each chain, as
well as the temperature ladder used to sample across chains, as sampling progresses. A key feature of the adjustment process
is that the maximum allowed change to any chain property diminishes over time, made inversely proportional to the number of
samples; this is necessary to ensure that the chains converge to the correct distribution in the limit of large numbers of samples
(Roberts and Rosenthal, 2007). The Obsidian implementation of PTMCMC also allows it to be run on distributed computing
clusters, making it truly parallel in resource use as well as in the requirement for multiple chains.

## 2.5 The original Moomba inversion problem

The goal of the original Moomba inversion problem (Beardsmore et al., 2016; McCalman et al., 2014a) was to identify potential
geothermal energy applications from hot granites in the South Australian part of the Cooper Basin (cf. Carr et al. (2016) for
a recent review of the Cooper Basin). Modeling the structure of granite intrusions and their temperature enabled the inference
of the probability of the presence of granite above 270 $^{\circ}$C at any point within the volume. The chosen region was a portion of
the Moomba gas field with dimensions of $35 \times 35 \times 12$ km volume centered at $-28.1^{\circ}$ S, $140.2^{\circ}$ E. The volume is divided
into six layers, with the first four being thin, sub-horizontal, Permo–Triassic sedimentary layers, the fifth corresponding to
Carboniferous–Permian granitoid intrusions (Big Lake Suite), and the sixth to a Proterozoic basement (Carr et al., 2016).
The number of layers and the priors on mean depths of layer boundaries were related to interpretations of depth-converted
seismic reflection horizons published by the Department of State Development (DSD) in South Australia (Beardsmore et al.,
2016). Data used in the inversion include Bouguer anomaly; total magnetic intensity; magnetotelluric sensor data; temperature
measurements from gas wells; and petrophysical laboratory measurements based on 115 core samples from holes drilled
throughout the region. Rock properties measured for each sample include density, magnetic susceptibility, thermal conductivity,
thermal productivity, and resistivity.

The original choices of how to partition knowledge between prior and likelihood struck a balance between accuracy of the
world representation and computational efficiency. The empirical covariances of the petrophysical sample measurements for
each layer were used to specify a multivariate Gaussian prior on that layer's rock properties; although these measurements
could be construed as data, the simplifying assumption of spatially constant mean rock properties left little reason to write
their properties into the likelihood. The gravity, magnetic, magnetotelluric, and thermal data all directly constrained rock
properties relevant to the geothermal application and were explicitly forward-modeled as data. "Contact points" from drilled
wells, directly constraining the layer depths in the neighborhood of a drilled hole as part of the likelihood, were available and
used to inform the prior, but not treated as sensors in the likelihood. Treating the seismic measurements as data would have
dramatically increased computational overhead relative to the use of interpreted reflection horizons as mean functions for layer



boundary depths in the prior. Using interpreted seismic data to inform the mean functions of the layer boundary priors also reduced the dimension of the parameter space, letting the control points specify long-wavelength deviations from seismically derived prior knowledge: each reflection horizon was interpolated onto a $20 \times 20$ grid, meaning that 400 control points per layer (resulting in 2400 parameters for the world geometry alone) would have been required to define the high-resolution reference

world.

Given this knowledge of the local geology (Carr et al., 2016; McCalman et al., 2014b), the world parameters for geometry were chosen as follows: The surface was fixed by a level plane at zero depth. The control point grids for the relatively simple sedimentary layers were specified by $2 \times 2$ grids of control points (lateral spacing: 17.5 km). The layer boundary for the granite intrusion layer used a $7 \times 7$ grid (lateral spacing: 5 km), and also underwent a nonlinear transformation stretching the boundary

vertically, to better represent the elongated shapes of the intrusions. Including the rock properties, this allowed the entire world to be specified by a vector of 101 parameters, a large but not unmanageable number.

Figure 3 show horizontal slices through the posterior probability density for granite at a depth of 3.5 km, similar to that shown in figure 9 of Beardsmore et al. (2016), for three MCMC runs sampling from the original problem. While the posterior samples from the previous inference are not available for quantitative comparison, we see reasonable qualitative agreement

with previous results in the cross-sectional shape of the granite intrusion.

## 3  Experiments

To demonstrate the impact of problem setup and proposal efficiency in a Bayesian MCMC scheme for geophysical inversion, we run a series of experiments altering the prior, likelihood, and proposal for the Moomba problem. We approach this variation as an iterative investigation into the nature of the data and the posterior's dependence on them, motivating each choice with the

intent of relating our findings to related 3-D inversion problems.

The datasets we use for our experiments are the gravity anomaly, total magnetic intensity, and magnetotelluric readings originally distributed as an example Moomba configuration with v0.1.1 of the Obsidian source code. In order to focus on information that may be available in an exploration context (i.e. publicly available geophysical surveys without contact points), we omit the thermal sensor readings, relying on a joint inversion of gravity, magnetic, and magnetotelluric data.

We run Obsidian's parallel-tempered sampler using 4 simultaneous temperature ladders or "stacks" of chains, each with 8 temperatures, as a baseline configuration. The posterior is formally defined in terms of samples over the world parameters, so when quantifying predictions for particular regions of the world and their uncertainty (such as entropy), the parameter samples are each used to create a voxelised realization of the 3-D world, and the average observable calculated over these voxelised samples. A quantitative summary of our results is shown in Table 1, including, for each run:

– the shortest ($\tau_{\min}$), median ($\tau_{\text{med}}$), and longest ($\tau_{\max}$) autocorrelation time measured for individual model parameters;

– the standard deviations $\sigma_{\text{grav}}$ and $\sigma_{\text{mag}}$, of the gravity and magnetic anomaly sensor data from the posterior mean forward model prediction, in physical units;





**Table 1.** Performance metrics for each run, including: best-case, median, and worst-case autocorrelation times for model parameters; standard deviations of residuals from the data for each sensor; volume-average information entropy; number of chain iterates; and CPU-hours per autocorrelation time.

| Run | $\tau_{i,\min}$ (/1000) | $\tau_{\mathrm{med}}$ (/1000) | $\tau_{i,\max}$ (/1000) | $\sigma_{\mathrm{grav}}$ (mgal) | $\sigma_{\mathrm{mag}}$ (nT) | $\bar{S}$ (bits) | $N$ | CPU (h) $/\tau_{\max}$ | Comments |
|---|---|---|---|---|---|---|---|---|---|
| A | 4.3 | 16.4 | 67.8 | 0.4 | 19.2 | 0.79 | 764.5k | 10.8 | baseline iGRW |
| A1 | 4.7 | 10.7 | 42.8 | 0.4 | 18.5 | 0.68 | 1566.5k | 8.1 | …with $N_\beta = 12$ |
| B | 2.1 | 4.0 | 28.4 | 0.5 | 18.8 | 0.66 | 628.8k | 5.5 | baseline pCN |
| B1 | 2.4 | 4.4 | 24.3 | 0.5 | 20.5 | 0.62 | 1166.5k | 6.2 | …with $N_\beta = 12$ |
| C | 1.9 | 17.4 | > 143.2 | 0.5 | 20.9 | 0.57 | 872.6k | > 19.7 | baseline aGRW |
| C1 | 2.7 | 14.1 | 310.6 | 0.4 | 17.1 | 0.61 | 2190.2k | 53.8 | …with $N_\beta = 12$ |
| D | 2.3 | 7.2 | 54.9 | 0.8 | 5.7 | 0.47 | 586.6k | 11.5 | Cauchy likelihood |
| E | 3.0 | 8.0 | > 172.1 | 0.7 | 6.4 | 0.51 | 669.2k | > 29.0 | 5 km margin |
| F1 | 12.0 | 101.9 | > 505.6 | 0.5 | 4.6 | 0.43 | 2386.4k | > 229.5 | smoothed data, $N_\beta = 12$ |
| F4 | 13.6 | 42.1 | 170.3 | 0.6 | 7.8 | 0.54 | 3823.7k | 39.9 | …subsampled to 100 pts/sensor |
| J | 1.6 | 26.3 | 115.4 | 0.8 | 7.0 | 0.61 | 1172.6k | 11.0 | loosen rock property priors |
| J2 | 2.1 | 7.9 | 53.8 | 1.1 | 9.4 | | 497.7k | 14.4 | …using 1 top layer only |
| K | 4.2 | 19.8 | 64.7 | 0.5 | 9.9 | 0.90 | 708.8k | 9.9 | loosen control point priors |
| K2 | 3.7 | 7.7 | 24.7 | 0.5 | 8.4 | | 479.1k | 7.4 | …using 1 top layer only |

- the mean information entropy $\bar{S}$ (Wellmann and Regenauer-Lieb, 2012) of the posterior probability density for granite, averaged over the volume beneath 3.5 km, in bits (i.e. presence or absence of granite; an entropy of 0 bits means total certainty, while 1 bit of entropy indicates total uncertainty);

- the CPU time spent per worst-case autocorrelation time, as a measure of computational efficiency.

## 3.1 Choice of within-chain proposal

The initial work of McCalman et al. (2014a) and Beardsmore et al. (2016) used an isotropic Gaussian random walk (iGRW) proposal within each chain,

$$\boldsymbol{\theta}' = \boldsymbol{\theta}_n + \eta\boldsymbol{u}, \qquad \boldsymbol{u} \sim N(\boldsymbol{0}, \boldsymbol{I}), \tag{14}$$

where $\eta$ is a (possibly adaptive) step size parameter. Each dimension of a sampled parameter vector is "whitened" by dividing it by a scale factor corresponding to the allowed full range of the variable (of order a few times the prior width; this also accounts for differences in physical units between parameters). This should at least provide a scale for the marginal distribution of each parameter, but does not account for potential correlations between parameters. The covariance matrix of the iGRW proposal





is a multiple of the identity matrix, so that on average, steps of identical extent are taken along every direction in parameter space. When tuning the proposal, the adaptive scheme tunes only an overall step size applying to all dimensions at once.

The iGRW proposal is the simplest proposal available, but as noted above, it loses efficiency in high-dimensional parameter spaces, and it is unable to adapt if the posterior is highly anisotropic — for example, if parameters are scaled inappropriately
or are highly correlated. The overall step size will adapt to the proposal width along the narrowest dimension, and the random walk will slowly diffuse along the other dimensions; the time it takes to traverse the entire posterior distribution should scale roughly as the square of the condition number of the Fisher matrix.

If the global shape of the posterior is not known, it can be determined using an adaptive/anisotropic Gaussian random walk (Haario et al., 2001). The covariance of the aGRW proposal is calculated in terms of the sample covariance of the chain history
$\{\boldsymbol{\theta}^{[j]}\}$:

$$\boldsymbol{\theta}' = \boldsymbol{\theta}_n + \eta \boldsymbol{u}, \qquad \boldsymbol{u} \sim N(\mathbf{0}, \boldsymbol{\Sigma}_n), \tag{15}$$

in which

$$\boldsymbol{\Sigma}_n = \frac{n}{n+a} \mathrm{cov}\left\{\boldsymbol{\theta}^{[j]}\right\} + \frac{a}{n+a} \boldsymbol{I}, \tag{16}$$

where $a$ is a timescale for adaptation (measured in samples). As the length $n$ of the chain increases, the proposal will smoothly
transition from an isotropic random walk to an anisotropic random walk with a covariance structure that reflects the chain history.

A third proposal, addressing high-dimensional parameter spaces, is the *preconditioned Crank-Nicholson* (pCN) proposal (Cotter et al., 2013):

$$\boldsymbol{\theta}_{n+1} = \sqrt{1-\eta^2}\boldsymbol{\theta}_n + \eta \boldsymbol{u}, \qquad \boldsymbol{u} \sim P(\boldsymbol{\theta}) \tag{17}$$

with $0 < \eta < 1$ and $P(\boldsymbol{\theta})$ a multivariate Gaussian prior. For $\eta \ll 1$, the proposal resembles a GRW proposal with small step size, while for $\eta \sim 1$ the proposal becomes a draw from the prior. This proposal results in a sampling efficiency that is independent of the dimensionality of $\boldsymbol{\theta}$; in fact, it was developed by Cotter et al. (2013) to sample infinite-dimensional function spaces, arising in inversion problems using differential equations as forward models, where the prior is specified in the eigenbasis for the forward model operator. In our case, the prior incorporates the correlation between neighboring control points in the
Gaussian process layer boundaries, so we might expect that a proposal that respects this structure would improve sampling.

Our first three runs (A, B, C) use the iGRW, pCN, and aGRW (with $a = 10$) proposals respectively. All three algorithms give roughly similar results on the baseline dataset. The autocorrelation time for this problem remains very long, of the order of $10^4$ samples. This means that $\sim 10^6$ samples are required to achieve reasonable statistical power.

There are nevertheless differences in efficiency among the samplers. The pCN proposal has not only the lowest median
autocorrelation, but the lowest worst-case autocorrelation across dimensions. The aGRW proposal has the largest spread in autocorrelation times across dimensions, with its median performance comparable to iGRW and its worst-case performance at least three times worse (it had still failed to converge after over 1000 CPU-hours). Repeat trials running for twice as many





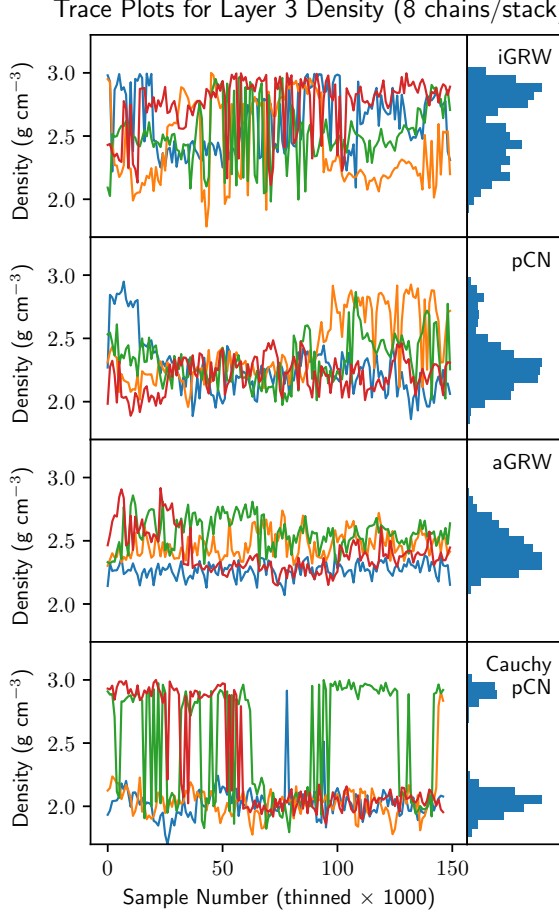

**Figure 2.** Trace plots (left) and marginal densities (right) for layer 3 rock density as explored by iGRW, pCN, and aGRW proposals, and by a pCN proposal under a Cauchy likelihood (top to bottom). The four colors represent the four different chains.

samples with 12 chains per stack instead of 8 (Runs A1, B1, C1) produced similar results, although we were then able to reliably measure the worst-case autocorrelation time for aGRW. For all samplers, but most noticeably aGRW, the step size can take a long time to adapt. Large differences are sometimes seen in the adapted step sizes between chains at similar temperatures in different stacks, and do not always increase monotonically with temperature.

5     The differences are shown in Fig. 2, showing the zero-temperature chains from the four stacks in each run sampling the marginal distribution of the rock density for layer 3, a bimodal parameter. The iGRW chains converge slowly, and though they manage to travel between modes with the help of parallel-tempered swap proposals, the relative weights of the two modes are not fully converged and vary between re-runs at a fixed length. Each aGRW chain has a relatively narrow variance and none successfully crosses over to the high-density mode despite parallel-tempered swaps. Only the pCN chains converge "quickly"

10    (after about 70k samples) and are able to explore the full width of the distribution.





These behaviors suggest that the local shape of the posterior varies across parameter space, so that proposals that depend on a global fixed scaling across all dimensions are unlikely to perform well. The clearly superior performance of pCN for this problem is nevertheless intriguing, since for sufficiently small step size near $\beta = 1$, the proposal reduces to GRW.

Figure 2 shows that iGRW and aGRW have more trouble traveling between different posterior modes than pCN. This is true despite the fact that all three proposals are embedded within a PTMCMC scheme with a relatively simple multivariate Gaussian prior, to which aGRW should be able to adapt readily. We believe pCN will prove to be a good baseline proposal for tempered sampling of high-dimensional problems because of its prior-preserving properties, which ensure peak performance when constraints from the data are weak. As the chain temperature increases, the tempered posterior density approaches the prior, so that pCN proposals with properly adapted step size will smoothly approach independent draws from the prior with an acceptance probability of 1. The result is that when used as the within-chain proposal in a high-dimensional PTMCMC algorithm, pCN proposals will result in near-optimal behavior for the highest-temperature chain, and should explore multiple modes much more easily than GRW proposals.

This behavior stands in contrast to GRW proposals, for which the acceptance fraction given any particular tuning will approach zero as the dimension increases. In fact, aGRW's attempt to adapt globally to proposals with local structure may mean mid-temperature chains become trapped in low-probability areas and break the diffusion of information down to lower temperatures from the prior. A more detailed study of the behavior of these proposals within tempered sampling schemes would be an interesting topic for future research.

## 3.2 Variations in likelihood / noise prior

In the fiducial Moomba configuration, the priors on the unknown variance of the Gaussian likelihood for each sensor are relatively informative. For example, the choice $\alpha = 5$, $\beta = 0.5$ — used for the gravity and magnetotelluric sensors — corresponds to noise with standard deviation $21\% < \sigma < 52\%$ with 95% probability, and a median of $\sigma = 32\%$ (that is, as a percentage of the sample standard deviation of the data in its original units). The resulting $t$-distribution for each data point has $\nu = 2\alpha = 10$ degrees of freedom, so that the non-Gaussian tails resulting from an imprecisely known noise variance are strongly suppressed. Thus the likelihood is close to being Gaussian with fixed variance 0.32. The magnetic sensor, on the other hand, uses $\alpha = 1.25$, $\beta = 1$, a much more permissive prior.

If specific informative prior knowledge about observational errors exists, using such a prior, or even fixing the noise level outright, makes sense. In cases where the amplitude of the noise term is not well-constrained, using a broader prior on the noise term may be preferable. When more than one sensor with unknown noise variance is used, identical broad priors allow the data to constrain the relative influence of each sensor on the final results. The trade-off is that a more permissive prior on the noise variance could mask structured residuals due to model inadequacy or non-Gaussian outliers in the true noise distribution.

The idea that such broad assumptions could deliver competitive results arises from the incorporation of Occam's razor into Bayesian reasoning, as demonstrated in Sambridge et al. (2012). For example, the log likelihood corresponding to independent





Gaussian noise is

$$\log \mathcal{L} = -\frac{1}{2} \sum_{j=1}^{N_d} \left[ \frac{(f_{sj}(\boldsymbol{\theta}) - \mathcal{D}_{sj})^2}{\sigma^2} + \log 2\pi\sigma^2 \right]. \tag{18}$$

Ordinary least-squares fitting maximizes the left-hand term inside the sum, and the right-hand term is a constant that can be ignored if the observational uncertainty $\sigma$ is known. This clearly penalizes worlds $\boldsymbol{\theta}$ resulting in large residuals. Suppose that
$\sigma$ is unknown, however, and is allowed to vary alongside $\boldsymbol{\theta}$: the left-hand term penalizes small (overly confident) values of $\sigma$, while the right-hand term penalizes large values of $\sigma$ corresponding to an assumption that the data are entirely explained by observational noise.

    Typical residuals from the fiducial inferences correspond to about 10% of the dataset's full range, so we next perform a run in which the noise prior is set to $\alpha = 0.5, \beta = 0.05$ for all samples. The corresponding likelihood (with the noise variance prior
integrated out) becomes a Cauchy (or $t_1$) distribution, with thick tails that allow substantial outliers from the core. This choice of $\alpha$ and $\beta$ thus also allows us to make contact with prior work where Cauchy distributions have been used (B.C Silva and Cutrim, 1989; de la Varga et al., 2018): a Gaussian likelihood with unknown, $\mathrm{IG}(0.5, \beta_s)$-distributed variance is mathematically equivalent to a Cauchy likelihood with known scale $2\beta_s$. The two choices are conceptually different, since in the Gaussian case outliers appear when the wrong variance scale is applied, whereas in the Cauchy case the scale is assumed known and the data
have an intrinsically heavy-tailed distribution.

    Under this new likelihood the residuals from the gravity observations increase (by about a factor of 1.5–2), while the residuals from the magnetic sensors decrease (by a factor of 3–4). This rebalancing of residuals among the sensors with an uninformative prior can be used to inform subsequent rounds of modelling more readily.

    The inference also changes: in run D, a granite bridge runs from the main outcrop to the eastern edge of the modelled volume,
with the presence of granite in the northwest corner being less certain. Agreement with run B and with the Beardsmore et al. (2016) map is still good along the eastern edge. The posterior entropy also decreases substantially, due to increase in the probability of granite structures at greater depths (beneath 3.5 km).

    The weight given to the gravity sensor is thus an important factor determining the behavior of the inversion throughout half the modeled volume. With weakened gravity constraints, the two modes for the inferred rock density in layer 3 separate widely
(see Fig. 2), though the algorithm is still able to move between the modes occasionally. The marginal distributions of the other rock properties do not change substantially, and remain unimodal.

    The comparison map for the inversion of Beardsmore et al. (2016) comes from the deterministic inversion of Meixner and Holgate (2009), which uses gravity as the main surface sensor but relies heavily on seismic data, with reflection horizons used to constrain the depth to basement, and measurements of wave velocities (which correlate with density) from a $P$-wave
refraction survey to constrain density at depth. While Meixner and Holgate (2009) mention constraints on rock densities, no mention is made of the level of agreement with the gravity data.

    Without more information — a seismic sensor in our inversion, priors based on the specific seismic interpretations of Meixner and Holgate (2009), or specific knowledge about the noise level in the gravity dataset that would justify an informative prior — it is hard to say how concerned we should be about the differences between the deterministic inversion and our probabilistic




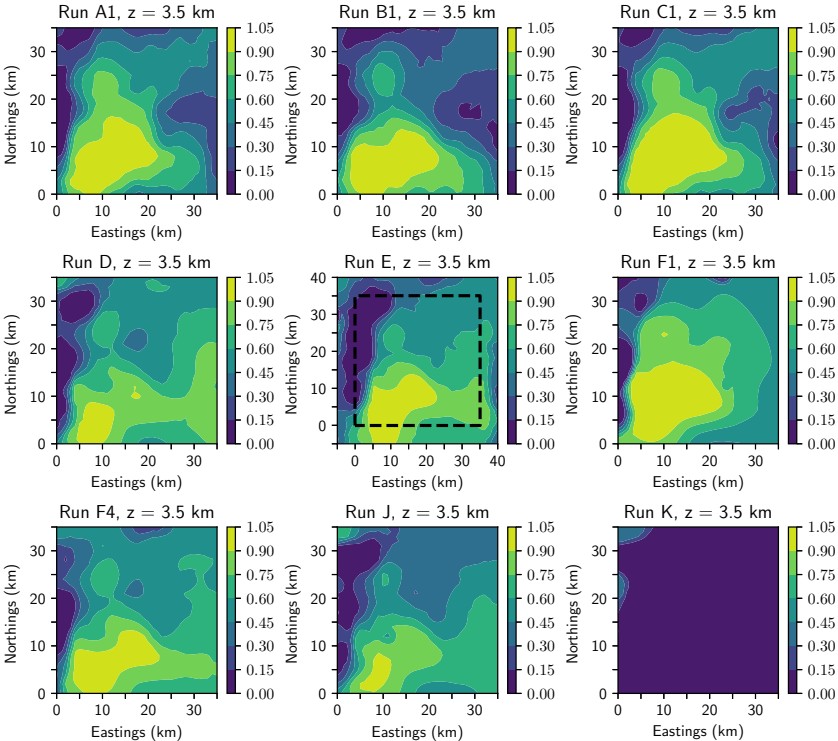

**Figure 3.** Slices through the voxelised posterior probability of occupancy by granite for each run at a depth of 3.5 km.

version. The comparison certainly highlights the potential importance of seismic data, both as a constraint on rock properties at depth and on the geometry of geological structures.

Indeed, one potential weakness of this approach to balancing sensors is model inadequacy: the residuals from the inference may systematic residuals from unresolved structure in the model, in addition to sensor noise. The presence of such residuals is a

5 model selection question that in a traditional inversion context would be resolved by comparing residuals to the assumed noise level, but this depends strongly upon informative prior knowledge of the sensing process for *all sensors* used in the inversion. The remaining experiments will use the Cauchy likelihood unless otherwise specified.

### 3.3 Boundary conditions

The boundary conditions Obsidian imposes on world voxelisations assume that rock properties rendered at a boundary edge

10 (north/south, east/west) extend indefinitely off the edges, e.g.

$$\rho_{is}(x < x_{\min}) = \rho_{is}(x_{\min}),$$
$$\rho_{is}(x > x_{\max}) = \rho_{is}(x_{\max}).$$



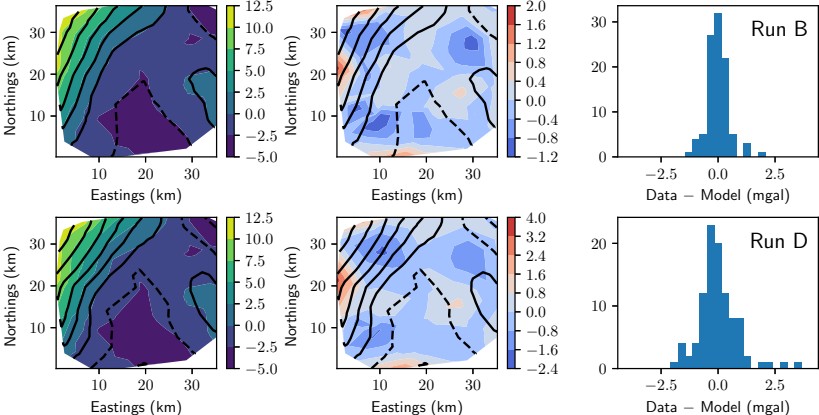

**Figure 4.** Gravity anomaly at the surface ($z = 0$). In a contour plot (left): filled contours = observations, black lines = mean posterior forward model prediction. Residuals of observations from the mean posterior forward model are also shown as a contour map (middle) and histogram (right).

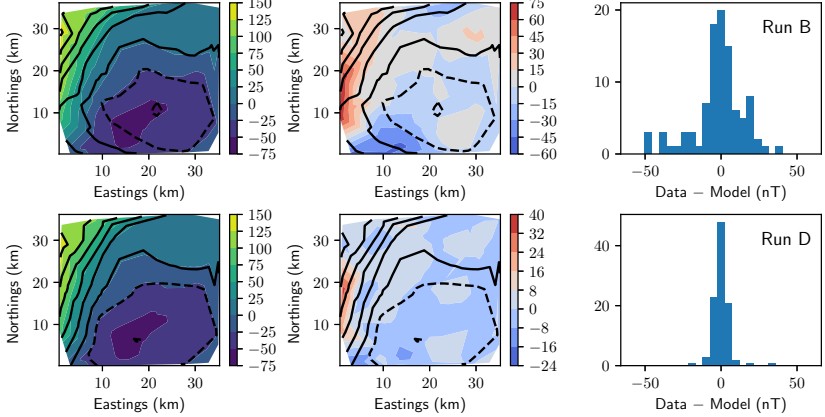

**Figure 5.** Magnetic anomaly at the surface ($z = 0$) in the same format as Fig. 4.

This may not be a good approximation when rock properties show strong gradients near the boundary. The residual plots shown in Fig. 4 and 5 show persistently high residuals along the western edge of the world, where such gradients appear in both the gravity anomaly and the magnetic anomaly.

For geophysical sensors with localized response, one way to mitigate this is to include in the world representation a larger area than the sensor data cover, incorporating a margin with width comparable to the scale of boundary artifacts, in order to let the model respond to edge effects for sensors with a finite area of response. In run E, we add a boundary zone of width 5 km around the margins of the world, while increasing the number of control points in the granite intrusion layer boundary from 49





($7 \times 7$ grid) to 64 ($8 \times 8$ grid). Neither the model residuals nor the inferred rock geometry substantially differs from the previous run, suggesting that the remaining outliers are actual outliers and not primarily due to mismatched boundary conditions. The autocorrelation time, however, increases substantially due both to the increase in the problem dimension and the fact that the new world parameters are relatively unconstrained, hence poorly scaled relative to the others.

## 3.4 Smoothed/resampled sensor data

The Obsidian likelihood assumes that the observational noise fluctuations in the sensor data are statistically independent. This is also the implicit assumption behind least-squares fitting, but it may not be true if the data have been interpolated, resampled, or otherwise modified from original point observations. For example, gravity anomaly and magnetic anomaly measurements are usually taken at ground level along access trails to a site, or along spaced flight lines in the case of aeromagnetics. In online data releases, the original measurements may then be interpolated or resampled onto a grid, changing the number and spacing of points and introducing correlations on spatial scales comparable to the scale of the smoothing kernel. This resampling of observations onto a regular grid may be useful for traditional inversions using Fourier transform techniques. However, if used uncritically in a Bayesian inversion context, correlations in residuals from the model may then arise from the resampling process rather than from model misfit, resulting in stronger penalties in the likelihood for what would otherwise be plausible worlds, and muddying questions around model inadequacy.

To simulate these effects, we interpolated the original, irregularly sampled gravity and magnetic anomaly data onto regular grids with 1.5 km spacing, resulting in 552 samples for each dataset (from the original observations, as there would be no way after the regridding to tell how many independent observations there had been). For each interpolation we used the maximum *a posteriori* fit of a Gaussian process regression with a square exponential kernel; the best-fit correlation length was 8 km (7 km) with residuals of about 4% (2%) of standard deviation for the gravity (magnetic) anomaly data. To the extent that these potential-field sensors represent moving averages of the underlying rock properties on some length scale, these results show why a $7 \times 7$ grid of control points (spacing 4.4 km) should provide adequate resolution for reconstruction of bulk geology at 3.5 km target depth. Although different interpolation schemes, such as linear, loess, or spline interpolation, may be applied for resampling and smoothing in contemporary online surveys, Gaussian process regression can be easily applied to irregularly sampled data and so forms a reasonable test case here.

The residuals from the original data are well below the typical residuals of model inadequacy based on other runs. However, the median and worst-case autocorrelation times are over 10 times longer than for the unsmoothed data of Run D, and indeed the run failed to converge before the adaptation decreased to negligible influence. The non-converged Run F posterior looks different, with a much higher probability of granite at 3.5 km depth than the other inversions, together with lower volume entropy in the inversion overall. The bridge characterizing Runs D and E (with the same likelihood) vanishes. Despite the superficial improvement in residuals, we might well view the results with suspicion, since information has been taken out of the data.

Variations on this standard case make little difference. Running with up to 16 parallel-tempered chains, thereby doubling the computation time, produces similar results. Making the noise prior more restrictive, in case the high level of correlation is due





to enabling exploration of too large a space (for example $\alpha = 10$, $\beta = 0.3$, a roughly Gaussian likelihood with fixed variance of 3%) results in higher MAP probability but no improvement in sampling. Adding random Gaussian noise of 3% to the smoothed data, to match the fluctuations around the original data and satisfy the assumptions of the noise prior, also has no obvious effect. Fitting a random subsample of 100 points each from the smoothed gravity and magnetic anomaly datasets, producing smoothed

datasets with the same size and approximate distribution as the original data (Run F4), reduces the autocorrelation time by more than half, but it remains six times as long as Run D.

     The reprocessing of the data has had several effects: First, it has increased the number of (assumed independent) data points to fit, tightening constraints where no new information has been added and making the posterior more difficult to explore. Second, even after subsampling the dataset to match the original number of points, some correlations still remain; the

probability that neighboring points will deviate from the fit in the same direction is larger than for the unsmoothed data. Thus, while spatially coherent residuals might ordinarily point to inadequacy in the world parametrization, the results are unclear if the data have been smoothed. A kernel smoothing radius of 7 km results in at most $(35\text{ km}/7\text{ km})^2 = 25$ independent spatial regions, so this is the effective sample size of our smoothed data. The loss of information means that not only is the answer more uncertain (even biased), but the algorithm mistakenly reports a *less* uncertain answer through the smaller posterior variance.

This cautionary tale shows that for best results, the input data should not be smoothed, or should at least be subsampled to reduce correlation between points (if the correlation scale is known). Improved results could also be obtained by using a multivariate Gaussian likelihood with correlations on the appropriate spatial scale (that is, a Gaussian process likelihood). A Gaussian process likelihood, however, complicates matters by introducing a length scale hyperparameter into the sampling, and by risking confusion between spatially coherent model errors and correlated observational noise.

**3.5   Looser priors on rock properties and layer depths**

In cases where samples of rock for a given layer are few or unavailable, the empirical covariance used to build the prior on rock properties may be highly uncertain or undefined. In these cases, the user may have to resort to a broad prior on rock properties. The limiting case is when no petrophysical data are available at all. Similarly, definitive data on layer depths may become unavailable in the absence of drill cores, or at least seismic data, so that a broad prior on control point depths may also become

necessary.

     We re-run the main Moomba analysis using two new priors. The first (run J) simulates the absence of petrophysical measurements. The layer depth priors are the same as the fiducial setup, but the rock property prior for each layer is now replaced by an independent Gaussian prior on each rock property, with the same mean as in previous runs but a large width common to all layers:

$$\rho_{is} \sim \mathcal{N}(\mu_{\rho_{is}}, \sigma_{\rho_{is}}). \tag{19}$$

The standard deviations are 0.2 g cm$^{-3}$ (density), 0.5 (log magnetic susceptibility), and 0.7 (log resistivity in $\Omega$ m).

     The Run J voxelisation shows reasonable correspondence with the baseline run D, though with larger uncertainty, particularly in the northwest corner. In the absence of petrophysical samples, but taking advantage of priors on overlying structure from



seismic interpretations, a preliminary segmentation of granite from basement can thus still be obtained using broad priors on rock properties. Although the algorithm cannot reliably infer the bulk rock properties in the layers, the global prior on structure is enough for it to pick out the shapes of intrusions by looking for *changes* in bulk properties between layers.

The second run (run K) removes structural prior information instead of petrophysical prior information. The priors on rock properties are as in the fiducial setup, but the control point prior for each layer is replaced by a multivariate Gaussian with the same anisotropic Gaussian covariance,

$$\mathbf{\Sigma_\alpha} = \sigma_\alpha \begin{bmatrix} 1.0 & 0.5 & \dots & 0.5 \\ 0.5 & 1.0 & \dots & 0.5 \\ \vdots & \vdots & \ddots & \vdots \\ 0.5 & 0.5 & \dots & 1.0 \end{bmatrix} \tag{20}$$

with $\sigma_\alpha = 3$ km.

Run K yields no reliable information about the location of granite at 3.5 km depth. This seems to be due solely to the uncertain thickness of layers of sedimentary rock that are constrained to be nearly uniform horizontal slabs in Run J, corresponding to a known insensitivity to depth among potential-field sensors. When relaxed, these strong priors cause a crisis of identifiability for the resulting models. Further variations on Runs J and K show that replacing these multiple thin layers with a single uniform slab of $\sim 3$ km depth (Runs J2 and K2) does not aid either convergence or accuracy, as long as more than one layer boundary is allowed to have large-scale structure.

As mentioned above and in Beardsmore et al. (2016), the strong priors on layer boundaries and locations were originally derived from seismic sensor data. Such data will not always be available, but seem to be critical to constrain the geometry of existing layers to achieve a plausible inversion at depth.

## 4 Discussion

The clearest lesson we can draw from the various inversions we have run is that the posterior uncertainty can be much larger than one might expect from point-estimate or deterministic inversions. Our results were sensitive to the MCMC proposal used (in that some proposals were extremely inefficient and gave wrong results if stopped early; see Fig. 2); to the assumed weighting given to different sensors; to the way in which data might have been pre-processed; and to the quality and quantity of informative prior information.

The changes in the posterior under different priors are not always intuitive: unrealistically tight constraints can hamper sampling, but relaxing priors may sometimes widen the separation between modes (as shown in Fig. 2), which also makes the posterior difficult to sample. Additionally, particular weaknesses in sensors, such as the insensitivity of potential-field sensors to the depth of geological features or to the addition of any horizontally invariant density distribution, can make it impossible to distinguish using those data between multiple plausible alternatives, adding to the irregularity and multi-modality of the posterior.





While any single data source may be easy to understand on its own, unexpected interactions between parameters can also arise. Structural priors from seismic data or geological field measurements appear to play a crucial role in stabilizing the inversions in this paper, as seen by the collapse of our inversion after relaxing them.

Our findings reinforce the impression that to make Bayesian inversion techniques useful in this context, the computational burden must be reduced by developing efficient sampling methods. Three complementary ways forward present themselves:

1. to develop MCMC proposals, or non-parametric methods to approximate probability distributions, that both function in (relatively) high-dimensional spaces and capture local structure in the posterior;

2. to develop fast approximate forward models for complex sensors (especially seismic) that deliver detailed information at depth, along with new ways of assessing and reducing model inadequacy;

3. to develop richer world parametrizations of 3-D geological models that faithfully represent real-world structure in as few dimensions as possible.

All three of the MCMC proposals studied here are variations of random walks, which explore parameter space by diffusion and do not easily handle posteriors with detailed local covariance structure such as the ones we find here. Proposals that can sense and adjust to local structure from the present state require, almost by definition, knowledge of gradients (Neal et al., 2011) or higher-order curvature tensors (Girolami and Calderhead, 2011), which in turn require gradients of both the prior and the likelihood (in particular, of forward models).

Forming gradients of forward models by finite differences is likely to be as prohibitively expensive as not having gradients; furthermore, practitioners may not have the luxury of rewriting their forward model code to return gradients. This is one goal of writing fast emulations of forward models, particularly emulations for which derivatives can be calculated analytically (see for example Fichtner et al., 2006a, b). Smooth universal approximators, such as artificial neural networks, are one possibility; Gaussian process latent variable models (Titsias and Lawrence, 2010) or Gaussian process regression networks (Wilson et al., 2012) are others, which would also enable nonlinear dimensionality reduction for difficult forward models or posteriors. reduction. Algorithms that alternate between fast/approximate forward models for local exploration, on the one hand, and expensive/precise forward models for evaluation of the objective function, on the other, have proved useful in engineering design problems Jin (2011); Sóbester et al. (2014). These approximate emulators give rise to model inadequacy terms in the likelihood, which can be explicitly addressed; for example, Köpke et al. (2018) present a geophysics inversion framework in which the inference scheme learns a model inadequacy term as sampling proceeds, showing proof of principle on a crosshole georadar tomography inversion. A related, complementary route is to produce analytically differentiable approximations to the posterior, built as the chain explores the space (Strathmann et al., 2015; Lan et al., 2016).

Another source of overall model inadequacy comes from the world parametrization which can be viewed as part of the prior. Obsidian is tuned to match sedimentary basins; its world parametrization is too simple to represent more complex structures, particularly those with abrupt variations caused, for example, by fault displacement. The GemPy package developed by de la Varga et al. (2018) makes an excellent start on a more general-purpose package. GemPy is also specifically written to take advantage of autodifferentiation, providing ready gradient information for the prior.



## 5   Conclusions

We have performed a suite of 3-D Bayesian geophysical inversions for the presence of granite at depth in the Moomba gas field of the Cooper basin, altering aspects of the problem setup to determine their effects on the efficiency and accuracy of MCMC sampling. Our main findings are as follows:

  – Parametrized worlds have much lower dimensionality than non-parametric worlds, and the parameters also offer a more interpretable description of the world — for example, boundaries between geological units are explicitly represented. However, the resulting posterior has complex local covariance structure in parameter space, even for linear sensors.

  – Although isotropic random walk proposals explore such posteriors inefficiently, poorly adapted anisotropic random walks are even less efficient. A modified high-dimensional random walk such as pCN outperforms these proposals, and
the prior-preserving properties of pCN make it especially attractive for use in tempered sampling. However, proposals using gradients from autodifferentiation are probably needed to make further progress in this area.

  – The shape of the posterior and number of modes can also depend in complex ways upon the prior, making tempered proposals essential.

  – In cases where the relative observational noise levels in the data are not well-constrained, using identical, uninformative
priors on the noise level for each sensor allows the inversion algorithm to rebalance information among sensors for a better fit.

  – Smoothing or resampling sensor data leads to loss of sampling efficiency as well as a biased, unrealistically certain posterior. In these cases, subsampling the data to reduce correlations can aid sampling. The introduction of correlations into the likelihood may also improve the accuracy of the posterior, although sampling may still be inefficient without a
properly tuned MCMC proposal.

  – Useful information about structures at depth can sometimes be obtained through sensor fusion even in the absence of informative priors. However, direct constraints on 3-D geometry from seismic interpretations or structural measurements seem to play a privileged role among priors, owing to the relatively weak constraints on depth of structure afforded by potential field methods.

In summary, both advanced MCMC methods and careful attention to the properties of the data are necessary for inversions to succeed.

*Code and data availability.* The code for version 0.1.2 of Obsidian is available at https://github.com/rscalzo/obsidian/tree/0.1.2-beta. All configuration files for 3-D model runs specified in this paper, together with corresponding datasets, are available in named subfolders of https://github.com/rscalzo/obsidian/tree/0.1.2-beta/examples/scalzo18, and are also provided as supplementary material.



*Author contributions.* The study was conceptualized by SC, who with MG provided funding and resources. RS was responsible for project administration and designed the methodology under supervision from SC and GH. RS and DK carried out development of the Obsidian code resulting in v0.1.2, carried out the main investigation and formal analysis, and validated and visualized the results. RS wrote the original draft text, of which all co-authors provided review and critical evaluation.

5  *Competing interests.* The authors declare that they have no conflict of interest.

*Acknowledgements.* This work is part of the Lloyd's Register Foundation – Alan Turing Institute Programme for Data-Centric Engineering. RS thanks Lachlan McCalman, Simon O'Callaghan, and Alistair Reid for useful discussions about the development of Obsidian up to v0.1.1.



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
