# Peer review of "Efficiency and robustness in Monte Carlo sampling 3-D geophysical inversions with Obsidian v0.1.2: Setting up for success"

_Geoscientific Model Development, 2018_

## Referee Comment (RC1) · Anonymous Referee #1 · 10 Apr 2019

General comments:

I have read the manuscript with interest and I think that it will be a good contribution to the field of integrated geophysical modelling and inversion. The manuscript is well written and well organized. The authors present an inversion code relying on Monte-Carlo sampling in a Bayesian framework. The theoretical background pertaining to the Parallel Tempered Markov-Chain Monte-Carlo (PTMCMC) that is provided allows a good understanding of the principles behind the implementation. The code they use is an extension of an existing software, and there is therefore not much information, for instance, about the way they calculate the forward geophysical problem. The manuscript

is relatively de-attached from the software the authors introduce, which allows it to remain general and to provide a good introduction to Bayesian and Monte-Carlo techniques. However, I think that it is a little bit too detached from the code itself and more indications as to how users could use Obsidian in practice and to reproduce the work presented would be useful. The example they use to illustrate the methodology is appropriate.

The literature is generally well reviewed and well used but there are a few occurrences where references are miscited or should be added (in particular when it comes to less statistical and more geological considerations). I come back to it where necessary in the detailed comments below. This paper is used as a companion paper by Olierook et al. (2019) and is cited multiple times by them. The authors should consider citing Olierook et al. (2019) as an application example.

An aspect which is practically missing from the manuscript relates to the computational requirements of inverse modelling using Obsidian v.0.1.2. The model the authors are using as an illustration example seems small and yet I have the impression that carrying out the inverse modelling was relatively computationally intensive. A little bit more information would be welcome, and it would be useful to geoscientists planning to use Obsidian v.0.1.2.

Does the implementation restrict the modelling of one given property (say, density contrast) to one type of sensors (say, gravimeters)? I am asking this question because of the way equation 9 is formulated. It seems to imply that one physical property cannot be recovered from the joint inversion of two datasets. For instance, this would mean that, in its current version, Obsidian would not support an extension to the joint inversion of gravity anomaly measurements with tensor gravity gradiometry to recover density contrast?

Detailed comments:

The comments below follow a linear progression of the manuscript. 'P' indicates page

number and 'l' indicates line number.

Title. 'Sampling of [. . .] inversions'. I think that you cannot sample an inversion as it is a process, but that you can do sampling for 3-D inversions.

P2.l2-3. several works have recognised the issue. Consider adding a few references. P2.l6-7. 'gravity, magnetic, and electrical measurements integrate data from the surrounding volume'. This is true for all geophysical methods, even high-frequency seismics. You can replace by something like 'All geophysical measurements [. . .].'

P2.l13-14: In the work of A. Tarantola, non-uniqueness is clearly stated. It is one of the limitations of geophysical inversion and mitigating it is one of the motivations for integration and joint inversion as presented in this manuscript. Consider adding a word about non-uniqueness in geophysics to this sentence and perhaps another reference (for instance Sambridge (1998) might be relevant here).

P2.l22. 'All input sources of information [. . .] are probability distributions'. This is not the case in all inversion schemes. If this is a general truth you are saying (and I think it is a general truth), and if this is how all inputs are treated in your work/Obsidian, then consider stating it clearly.

P3.l3-4. 'posterior around each local maximum may in these cases significantly underestimate uncertainties'. This is a good point and it is often overlooked. Consider adding a reference to support this or an example illustrating this.

P3.l10-11. 'Giraud et al. (2017, 2018) demonstrate an optimization-based Bayesian inversion framework for 3-D geological models, which finds the maximum of the posterior distribution (maximum a posteriori, or MAP), and expresses uncertainty in terms of the posterior covariance around the MAP solution; while they show that fusing data reduces uncertainty around this mode, they do not attempt to find or characterize other modes, or higher moments of the posterior'. This is partially true. Giraud et al. (2017, 2018) use uncertainty information and assess the reduction of uncertainty after inversion, and find the maximum of the MAP, but they do not show the posterior explicitely. Giraud et al. (2016) on the other hand, do calculate the posterior covariance matrix.

P6.l9. The shape of matrix $\Sigma$ might be determined using probabilistic geological modelling (e.g, Wellmann et al. (2010), Pakyuz-Charrier et al. (2018a), de la Varga et al. (2018)). You could add that in the discussion.

P6.l16. "These authors found that in general updating blocks of parameters simultaneously was inefficient". My impression is that you also refer to inversions schemes using graphic cuts to update the models. If this is the case please state it clearly/briefly.

P6.l31. "using information from ensembles of particles", does the comment also extend to inversions using particle swarm optimization? If this is the case please state it clearly/briefly

P6.l26-30. You could consider making it clearer that this is what your version of Obsidian does so that readers/users are not wondering.

P8.l22. The acronym 'IACT' is used only in this place. Please remove.

P9.l29. I think that the usage of the word 'layer' is a bit confusing from a geological point of view as you later on refer to as an inclusion as a layer, which it is not. Please use more appropriate vocabulary.

P10.l8. Maybe you can state later in the manuscript that your implementation of geological structures is more suited to basin scenarii (and therefore oil and gas exploration cases), and that in hard rock / mining scenarii, different geological modelling approaches can be followed (as you do near the end of the manuscript when refering to gempy).

P10.l13. Equation 14. Just for the sake of completeness you may consider to specify what x' and y' are.

P10.l16. Typo: the bracket needs to be removed.

P10.l25. Equation 12. Consider adding a short appendix detailing how it is derived.

P11.l34. – P12.l1. Note that drillhole uncertainty for control points can be modelled (Pakyuz-Charrier et al. (2018b)), as can seismic interpretation (Bond (2015), Schaaf and Bond (2019), Alcalde et al. (2017)).

P12.l4-5. I find this discretization a bit coarse. Is it because of the computation cost involved in PTMCMC or due to lack of information or to shorten run time?

P13.l1. Information entropy has been used in the geosciences after Wellmann and Regenauer-Lieb (2012) introduced it to the field, but it was initially introduced by Shannon (1948). Consider adding this reference, and possibly a brief statement explaining why it is appropriate to use it.

P14.l5-7. Please make this paragraph clearer.

P14.l32. The information about the number of computational hours is relevant only if the specs of the computer used are known. I think that more information about this aspect of the work presented and of Obsidian should be given: does it run on supercomputers, do it scale well? Just a little bit of information on this aspect would be useful to users and would strengthen the paper.

P16.l19-25. This paragraph is not very clear to me.

P17.l17: "Suppose that $\sigma$ is unknown, however, and is allowed to vary alongside [theta]" does it mean you allow heteroscedasticity? If so this needs to be stated.

P17.l8. I'm not sure I understand the usage of the term 'fiducial' here.

P18.l3. "one potential weakness of this approach to balancing sensors". How would that relate to defining the relative weight of the different types of sensors in the joint inversion problem?

P18.l10. Equation number is missing.

P19. The models are shown only in 2D. A 3D view would be welcome.

P20.l24-25. You mention a number of interpolation techniques. Have you tried kriging, as it is widely used in geostatistics?

P23.l16-17. I am not sure that I understand the meaning of this sentence. Please clarify. By gradients, do you refer to the jacobian matrix? Or am I missing something?

P23.l30. the package proposed by de la Varga et al. (2018) offers the advantage of being open source but it is not the only one performing probabilistic geological modelling. For instance, other works using ideas introduced by Wellmann et al. (2010) such as Pakyuz-Charrier et al. (2018a) also achieves this.

References:

Alcalde, J., C. E. Bond, G. Johnson, J. F. Ellis, and R. W. H. Butler, 2017, Impact of seismic image quality on fault interpretation uncertainty: GSA Today.

Bond, C. E., 2015, Uncertainty in structural interpretation: Lessons to be learnt: Journal of Structural Geology, 74, 185–200.

Giraud, J., V. Ogarko, M. Lindsay, E. Pakyuz-charrier, M. Jessell, R. Martin, E. Targeting, E. Targeting, E. Sciences, E. Targeting, E. Targeting, E. Sciences, and E. Targeting, 2019, Sensitivity of constrained joint inversions to geological and petrophysical input data uncertainties with posterior geological analysis: Geophysical Journal International, Accepted, Accepted, to appear.

Giraud, J., M. Jessell, M. Lindsay, E. Parkyuz-Charrier, and R. Martin, 2016, Integrated geophysical joint inversion using petrophysical constraints and geological modelling: SEG Technical Program Expanded Abstracts 2016, 1597–1601.

de la Varga, M., A. Schaaf, and F. Wellmann, 2018, GemPy 1.0: open-source stochastic geological modeling and inversion: Geoscientific Model Development Discussions, 1–50.

Olierook, H. K. H., R. Scalzo, D. Kohn, R. Chandra, E. Farahbakhsh, G. Houseman, C. Clark, S. M. Reddy, and R. D. Müller, 2019, Bayesian geological and geophysical data fusion for the construction and uncertainty quantification of 3D geological models: Solid Earth Discussions, 1–34.

Pakyuz-Charrier, E., M. Lindsay, V. Ogarko, J. Giraud, and M. Jessell, 2018a, Monte Carlo simulation for uncertainty estimation on structural data in implicit 3-D geological modeling, a guide for disturbance distribution selection and parameterization: Solid Earth, 9, 385–402.

Pakyuz-Charrier, E., J. Giraud, V. Ogarko, M. Lindsay, and M. Jessell, 2018b, Drillhole uncertainty propagation for three-dimensional geological modeling using Monte Carlo: Tectonophysics.

Sambridge, M., 1998, Exploring multidimensional landscapes without a map: Inverse Problems, 14, 427–440.

Schaaf, A., and C. E. Bond, 2019, Quantification of uncertainty in 3-D seismic interpretation: implications for deterministic and stochastic geomodelling and machine learning: Solid Earth Discussions, 1–18. Shannon, C. E. E., 1948, A Mathematical Theory of Communication: Bell System Technical Journal, 27, 379–423.

Wellmann, J. F., and K. Regenauer-Lieb, 2012, Uncertainties have a meaning: Information entropy as a quality measure for 3-D geological models: Tectonophysics, 526–529, 207–216.

Wellmann, J. F., F. G. Horowitz, E. Schill, and K. Regenauer-Lieb, 2010, Towards incorporating uncertainty of structural data in 3D geological inversion: Tectonophysics, 490, 141–151.
* * *

---

## Referee Comment (RC2) · Anonymous Referee #2 · 18 Apr 2019

GENERAL COMMENTS:

This study explores the influence of various practitioner decisions on MCMC posterior sampler efficiency for a geophysical joint inversion with a layered paramitrization; specifically, the influence several of proposal, prior, and likelihood function options. The tests are well designed and succeed in addressing the questions asked. I personally did not find much of the results and conclusions surprising, most of it could be deduced from purely theoretical grounds. However, the topic is important and this paper gives a good empirical basis from which future geophysical posterior sampling work can draw. On these grounds, I think it deserves to be published.

Most of the paper is well written and clear, with the introduction being the exception. It seems rushed and the odd use of Bayesian/statistical/probabilistic terminology (in the introduction only) suggests a lack of familiarity.

I personally don't see the need for additional detail on the software use and implementation since that is not what the paper is about and if anything the scope should be more contained not expanded.

The review of the MCMC literature review is extensive and was interesting to read. Complexity of the shape of the posterior is discussed several times but seldom in the context of previous work. For example, the non-linearity and complex correlations of physical parameters for non-unique magnetotelluric inversion is well known, but no overview is given here on that. I believe the discussion section could be improved by relating more to the known properties of the different geophysical forward problems.

I think the discussion section is needlessly bloated. Here the authors go into detail on many topics which the experiments shown here had no bearing on. Various things that could be done or might work are listed here which are in no way related to what the study presented actually did. I strongly suggest rewriting this section to be more on topic.

SPECIFIC COMMENTS:

P1L12) What does "improve inversion results" refer to? Most readers would assume that it means a more accurate inversion. Since accuracy of results,compared to reality, is never quantified in this work, I don't see how this claim is backed up. One might argue that if true sensor noise levels are known, uninformative priors on them would only increase chances of their miss-estimation. Counterarguments based on model inadequacy could be raised of course, but these are not things that this study shed light on so please remove this claim.

P1L13-15) I do not see why this claim about using gradient information is in the abstract. The statement is probably true, but this study did not show anything new to support it.

P2L25) It's not clear what is meant by posterior ensembles being a 'gold-standard'.

P2L29) Online updating is not necessary or sufficient for optimal for decision-making; these are separate things. The only relevance I can see here is that it could speed up decision making.

P3L3) What about overestimating uncertainties?

P3L4) The "no 'one-size-fits-all' solution exists" comment is very important. Perhaps give the reader some direction by citing something (e.g. Wolpert et al., 1997, No free lunch theorems for optimization: IEEE transactions on evolutionary computation, 1, 67-82.)

P4L21-29) This paragraph should probably lead with the last sentence (lines 27-29). The parts about deterministic inversion reads like an odd tangent and I didn't see the relevance and purpose of it until a second read through.

P7) I did not pick up on the fact that all your tests use PTMCMC until the second read-through; this section should probably make that more explicit.

P12L1-5) Where the seismic lines used to inform the layer interface Gaussian process variogram?

P14L8) What is 'global posterior shape'? is it always defined?

P14) Could you have used the layer Gaussian process covariances directly to create a proposal function, it seems like that is what the CNp effectively does?

P16L4) "Figure 2 shows that iGRW and aGRW have more trouble travelling between different posterior modes than pCN" Tell us how you deduce this from that figure.

P17) Please mention why MT noise levels were fixed.

P21) Increasing the amount of data points by interpolation seems like a terrible idea. Why would anyone even attempt it? The observed effects should be obvious. If there are actual examples of Bayesian posterior analysis papers which do this, please cite one; otherwise, this seems like an odd and unnecessary test to include.

P21L16-19) I don't know about gravity and and magnetic, but Gaussian process likelihood functions have been used for MT and seismic MCMC. Relevant work should be cited here, E.g.:

Agostinetti, N. P., and A. Malinverno, 2010, Receiver function inversion by transdimensional Monte Carlo sampling: Geophysical Journal International, 181, 858–872.

Bodin, T., M. Sambridge, H. Tkalčić, P. Arroucau, K. Gallagher, and N. Rawlinson, 2012, Transdimensional inversion of receiver functions and surface wave dispersion: Journal of Geophysical Research: Solid Earth, 117.

Xiang, E., R. Guo, S. E. Dosso, J. Liu, H. Dong, and Z. Ren, 2018, Efficient hierarchical trans-dimensional Bayesian inversion of magnetotelluric data: Geophysical Journal International, 213, 1751–1767.

Also, there are ways to learn the correlation during sampling:

Steininger, G., J. Dettmer, S. E. Dosso, and C. W. Holland, 2013, Trans-dimensional joint inversion of seabed scattering and reflection data: The Journal of the Acoustical Society of America, 133, 1347–1357.

P22L19) "The clearest lesson we can draw ..." I'm not sure why this is the lesson you lead with, in the introduction it was stated as known; almost every MCMC application to geophysics show this and it was not among the questions that your tests were set up to answer.

P22L20) "Our results were sensitive to ..." Each point raised in this sentence will be true for for any difficult posterior sampling problem. This is not a new result and this sentence adds nothing to the manuscript.

P22L34) I don't agree with the claim that either of these outcomes are counter-intuitive. Tighter constraints lead to narrower local optima, hence more sampling is needed. Cauchy likelihood functions are more likely to give multi-modal posteriors than Gaussian likelihood functions, even for the most trivial problems (e.g. with just one parameter).

P23L6-10) None of these three dot-point listed statements were informed by the experiments presented in this manuscript. The claims are also obvious and well known.

P24L10) "However, proposals using gradients from auto-differentiation are probably needed to make further progress in this area." This claim, while probably true, is not really backed up by what is in the manuscript. Why is it listed as a conclusion?

P24L14) This is a trivial claim by itself. How can it help design future work. Will the better fit derived from uninformative priors lead to more accurate results in terms of uncertainty estimation. This ties in with my comment for P1L12.

P2417) Without some guiding principle for how to do the sub-sampling, this is not useful.

TECHNICAL CORRECTIONS:

P2L20) "..., but about uncertainties." Awkward use of terminology, a Bayesian probability is an uncertainty and an assumption. Assumptions are specified as uncertainties quantified by probability distributions.

P2L26) "The posterior distribution is a representation of all possible outcomes and hence provides an internal estimate of uncertainty." The world parameterization is the representation of all 'possible' outcomes. What does the 'internal estimate' mean? This sentence is incoherent.

P6L12) Spell out what SGR stands for here.

P6L26-27) Grammar mistake.

P13L1) Table 1, what is N? First it was iteration count, then number of layers, then what? Readers shouldn't have to fish through the past 12 pages to find out.

P18L4) Grammar mistake.
* * *

---

## Author Comment (AC1) · 14 Jun 2019

**Responses to Anonymous Referee 1**

I have read the manuscript with interest and I think that it will be a good contribution to the field of integrated geophysical modelling and inversion. The manuscript is well written and well organized. The authors present an inversion code relying on Monte-Carlo sampling in a Bayesian framework. The theoretical background pertaining to the Parallel Tempered Markov-Chain Monte-Carlo (PTMCMC) that is provided allows a good understanding of

the principles behind the implementation.

We thank the referee for their constructive review.

> The code they use is an extension of an existing software, and there is therefore not much information, for instance, about the way they calculate the forward geophysical problem. The manuscript is relatively de-attached from the software the authors introduce, which allows it to remain general and to provide a good introduction to Bayesian and Monte-Carlo techniques. However, I think that it is a little bit too detached from the code itself and more indications as to how users could use Obsidian in practice and to reproduce the work presented would be useful. The example they use to illustrate the methodology is appropriate.

We'll respond to individual suggestions below, but will just point out here that all of the configuration files and data sets we used to generate these solutions are available as part of the repository. We will review the documentation on the repository to ensure that the instructions for running Obsidian with these configurations are straightforward and can be followed without an expert knowledge of the code's inner workings.

> The literature is generally well reviewed and well used but there are a few occurrences where references are miscited or should be added (in particular when it comes to less statistical and more geological considerations). I come back to it where necessary in the detailed comments below.

We appreciate the suggestions of appropriate papers to cite where provided and have included them as applicable.

This paper is used as a companion paper by Olierook et al. (2019) and is cited multiple times by them. The authors should consider citing Olierook et al. (2019) as an application example.

We agree. The Olierook et al. paper was still in prep at the time we submitted this paper, which is the only reason it hasn't appeared here. It is also still in review at Solid Earth, but at least a reference to the discussion paper can be included here.

An aspect which is practically missing from the manuscript relates to the computational requirements of inverse modelling using Obsidian v.0.1.2. The model the authors are using as an illustration example seems small and yet I have the impression that carrying out the inverse modelling was relatively computationally intensive. A little bit more information would be welcome, and it would be useful to geoscientists planning to use Obsidian v.0.1.2.

In general the use of parallel-tempering MCMC is already very computationally intensive, and Obsidian was conceived as a code optimized to run on large distributed clusters such as AWS. Although we don't try to hide this – even our most efficient runs use a few CPU-hours per independent sample (see Table 1) – we agree that some additional wording about the computational cost, and what one obtains for having paid that cost, could benefit the paper. We have added a paragraph ("Since only samples...") describing this to section 2.2.

Does the implementation restrict the modelling of one given property (say, density contrast) to one type of sensors (say, gravimeters)? I am asking this question because of the way equation 9 is formulated. It seems to imply that one physical property cannot be recovered from the joint inversion of

two datasets. For instance, this would mean that, in its current version, Obsidian would not support an extension to the joint inversion of gravity anomaly measurements with tensor gravity gradiometry to recover density contrast?

We understand the problem with wording here, and clarify that each Obsidian sensor can in principle respond to any combination of rock properties, and so multiple sensors can respond to the same rock property if desired. If a forward model for a tensor gravity gradiometry sensor were included in Obsidian, nothing would prevent the user from combining it with gravity anomaly. We have revised the text before Equation 9 to refer to "K rock properties necessary and sufficient to evaluate the forward models for all relevant sensors."

Moreover, Obsidian allows the user to formulate multivariate Gaussian petrophysical priors that treat rock properties as correlated, for example between rock density and seismic wave speed – this would then allow different data sets to jointly constrain rock properties even if each one responded to only one rock property. We make this explicit now in the Priors section: "This allows the user to formulate priors that capture intrinsic covariances between rock properties, though of a somewhat simpler form than the petrophysical mixture model of Giraud et al (2017)."

Title. 'Sampling of [. . .] inversions'. I think that you cannot sample an inversion as it is a process, but that you can do sampling for 3-D inversions.

Changed to "...for 3-D geophysical inversions...".

P2.l2-3. several works have recognised the issue. Consider adding a few references.

We already have cited a number of references that seem relevant to us here, but have clarified in the text that "the issue" is "[the expense of] acquiring direct observations at depth".

> P2.l6-7. 'gravity, magnetic, and electrical measurements integrate data from the surrounding volume'. This is true for all geophysical methods, even high-frequency seis- mics. You can replace by something like 'All geophysical measurements [...].'

Replaced as suggested.

> P2.l13-14: In the work of A. Tarantola, non-uniqueness is clearly stated. It is one of the limitations of geophysical inversion and mitigating it is one of the motivations for integration and joint inversion as presented in this manuscript. Consider adding a word about non-uniqueness in geophysics to this sentence and perhaps another reference (for instance Sambridge (1998) might be relevant here).

We have added the Sambridge reference as requested.

> P2.l22. 'All input sources of information [. . .] are probability distributions'. This is not the case in all inversion schemes. If this is a general truth you are saying (and I think it is a general truth), and if this is how all inputs are treated in your work/Obsidian, then consider stating it clearly.

We agree that this isn't true of all inversion schemes, but by definition it is true of Bayesian schemes – the posterior depends only upon the likelihood and prior, which are probability distributions, although the way in which probability is expressed is quite

flexible. We have changed this sentence to read: "In a Bayesian approach, model elements are flexible but all statements about the fit of a model, either to data or to pre-existing expert knowledge, are expressed in terms of probability distributions."

> P3.l3-4. 'posterior around each local maximum may in these cases significantly un- derestimate uncertainties'. This is a good point and it is often overlooked. Consider adding a reference to support this or an example illustrating this.

We have rephrased to: "Use of the inverse Fisher information matrix to describe posterior uncertainty implicitly assumes a single multivariate Gaussian mode; for posteriors with multiple modes or significant non-Gaussian tails, the inverse Fisher information provides a lower bound on the posterior variance (Cramer 1946; Rao 1945) and may be a significant underestimate."

> P3.l10-11. 'Giraud et al. (2017, 2018) demonstrate an optimization-based Bayesian inversion framework for 3-D geological models, which finds the maximum of the pos- terior distribution (maximum a posteriori, or MAP), and expresses uncertainty in terms of the posterior covariance around the MAP solution; while they show that fusing data reduces uncertainty around this mode, they do not attempt to find or characterize other modes, or higher moments of the posterior'. This is partially true. Giraud et al. (2017, 2018) use uncertainty information and assess the reduction of uncertainty after inversion, and find the maximum of the MAP, but they do not show the posterior explicitly. Giraud et al. (2016) on the other hand, do calculate the posterior covariance matrix.

We agree with this characterization and this sentence now states that "...they do not attempt to find and characterize other modes, and only Giraud et al. (2016) calculate the posterior covariance."

P6.l9. The shape of matrix $\Sigma$ might be determined using probabilistic geological modelling (e.g, Wellmann et al. (2010), Pakyuz-Charrier et al. (2018a), de la Varga et al. (2018)). You could add that in the discussion.

This point gets to the heart of how what we are doing differs from previous probabilistic modeling frameworks from the above authors. The uncertainty-propagation framework works well for structural data because the model is a direct interpolation of the data. This would correspond in our case to sampling from the prior, and hence for $\Sigma$ to take the shape of the prior, making MCMC (as opposed to simple Monte Carlo sampling) unnecessary. The addition of likelihood components for geophysical data, however, may narrow and shift the posterior shape, making naive sampling less efficient and making the optimal proposal shape less intuitive. This was recognized by de la Varga Wellmann (2016) who recast the problem in terms of MCMC.

We have added a new paragraph in the Introduction to distinguish MCUE from MCMC methodologically, since the need for such distinction comes up again later. In this place in the text, we have also included the sentence: "If constraints from additional data are weak, $\Sigma$ could take the shape of the prior; if there are no other constraints, as in MCUE (Pakyuz-Charrier et al 2018a,b), sampling directly from the prior may be easier."

P6.l16. "These authors found that in general updating blocks of parameters simultaneously was inefficient". My impression is that you also refer to inversions schemes using graphic cuts to update the models. If this is the case please state it clearly/briefly.

No, this was not our intention.

P6.l31. "using information from ensembles of particles", does the comment also extend to inversions using particle swarm optimization? If this is the case please state it clearly/briefly

In this section we are discussing only (Metropolis-Hastings) MCMC sampling schemes, not optimization schemes or other particle-based sampling methods such as sequential Monte Carlo. The sentence now reads: "Many other types of proposals can be used in Metropolis-Hastings sampling schemes, using information from ensembles of particles (as distinct from particle swarm optimization or sequential Monte Carlo; Goodman Weare 2010)..."

P6.l26-30. You could consider making it clearer that this is what your version of Obsidian does so that readers/users are not wondering.

We have updated the text to read: "PTMCMC is a meta-method used by Obsidian for sampling..." and have also made explicit in section 2.4 what changes we made between v0.1.1 and v0.1.2 to support this and the Olierook et al 2019 paper.

P8.l22. The acronym 'IACT' is used only in this place. Please remove.

Fixed.

P9.l29. I think that the usage of the word 'layer' is a bit confusing from a geological point of view as you later on refer to as an inclusion as a layer, which it is not. Please use more appropriate vocabulary.

We have replaced all occurrences of the word "layer" in this context with "unit".

P10.l8. Maybe you can state later in the manuscript that your implementation of geological structures is more suited to basin scenarii (and therefore oil and gas exploration cases), and that in hard rock / mining scenarii, different geological modelling approaches can be followed (as you do near the end of the manuscript when refering to gempy).

[Figure]

We do already have such a sentence about Obsidian's world parametrization in the last section of the Discussion; we have added some words about specific applications, as suggested.

> P10.l13. Equation 14. Just for the sake of completeness you may consider to specify what x' and y' are.

The text now reads: "a radial basis function kernel to describe the correlation structure of the surface between two surface locations $(x, y)$ and $(x', y')$".

> P10.l16. Typo: the bracket needs to be removed.

Fixed.

> P10.l25. Equation 12. Consider adding a short appendix detailing how it is derived.

We now have included a derivation. The reviewer comments for Olierook et al 2019 made a similar request, but that paper is still under review; we have varied the wording accordingly.

> P11.l34. – P12.l1. Note that drillhole uncertainty for control points can be modelled (Pakyuz-Charrier et al. (2018b)), as can seismic interpretation (Bond (2015), Schaaf and Bond (2019), Alcalde et al. (2017)).

Noted. The frameworks in these papers seem to be about uncertainty propagation (like MCUE) and so could be used to elicit a prior on geological parameters in the context of fusion with geophysical data, as we do here.

[Figure]

P12.l4-5. I find this discretization a bit coarse. Is it because of the computation cost involved in PTMCMC or due to lack of information or to shorten run time?

Using a finer grid for the mean interpreted reflection horizon would probably not have made much difference to the computational efficiency; we use this grid here in order to reproduce the Beardsmore et al. setup. A finer grid of control points, however, would have dramatically increased the dimension of the problem.

P13.l1. Information entropy has been used in the geosciences after Wellmann and Regenauer-Lieb (2012) introduced it to the field, but it was initially introduced by Shannon (1948). Consider adding this reference, and possibly a brief statement explaining why it is appropriate to use it.

Cited, with the statement "this measure is appropriate to summarize posterior uncertainty in categorical predictions such as the type of rock".

P14.l5-7. Please make this paragraph clearer.

We have expanded this description somewhat: "To maintain the target acceptance rate, the adapted step size approaches the scale of the posterior's narrowest dimension, and the random walk will then slowly explore the other dimensions using this small step size. The time it takes for a random walk to cover a distance scales as the square of that distance, so we might expect the worst-case autocorrelation time for random-walk MCMC in a long, narrow mode to scale as the condition number of the covariance matrix for that mode."

P14.l32. The information about the number of computational hours is relevant only if the specs of the computer used are known. I think that more

information about this aspect of the work presented and of Obsidian should be given: does it run on supercomputers, do it scale well? Just a little bit of information on this aspect would be useful to users and would strengthen the paper.

These questions are addressed in McCalman et al. (2014), but we now point the reader to them by adding the following text to the beginning of section 2.4: "Obsidian was designed to run on large distributed architectures such as supercomputing clusters. McCalman et al. (2014) shows that the code scales well to large numbers of processors, by allowing individual MCMC chains to run in parallel and initiating communication between chains only when a PTMCMC swap proposal is initiated. The inversion of Beardsmore et al. (2016) was performed on Amazon Web Services using 160 cores."

We also include technical specifications of the Artemis cluster on which our experiments were run at the beginning of section 3, and state that each run used 32 cores for up to 8 hours of wall time, to provide typical end users with a better idea of the requirements.

P16.l19-25. This paragraph is not very clear to me.

This paragraph is really about the alpha parameter in the sensor noise prior and how it relates to certainty about the noise level. We have rewritten to emphasize this: "The uncertainty on the variance of a sensor is determined by the $\alpha$ parameter in that sensor's prior, with smaller $\alpha$ corresponding to more uncertainty. For example, the gravity and magnetotelluric sensors use a prior with $\alpha = 5$, so that the resulting t-distribution for model residuals in the likelihood has $\nu = 2\alpha = 10$ degrees of freedom. The magnetic anomaly sensor prior uses $\alpha = 1.25$, allowing a residual distribution with thick tails closer to a Cauchy distribution than a Gaussian."

P17.l17: "Suppose that $\sigma$ is unknown, however, and is allowed to vary
alongside [theta]" does it mean you allow heteroscedasticity? If so this
needs to be stated.

In this case, all we mean is that the overall scale of homoscedastic errors may not
be known, and hasn't been included as part of the dataset. We have made this more
specific: "Suppose that $\sigma$ is not perfectly known a priori, however (but is still assumed
to be the same for all points in a single dataset, and is allowed to vary..."

P17.l8. I'm not sure I understand the usage of the term 'fiducial' here.

We mean the original Moomba inversion presented in Beardsmore et al 2016, and have
now replaced occurrences of the word "fiducial" with a citation to this previous work.

P18.l3. "one potential weakness of this approach to balancing sensors".
How would that relate to defining the relative weight of the different types of
sensors in the joint inversion problem?

The point we attempt to raise in this paragraph is that if the data for a given sensor
have real variation beneath the scale of the basic world parametrization to resolve, that
variation will be treated by our approach as "noise".

P18.l10. Equation number is missing.

Fixed.

P19. The models are shown only in 2D. A 3D view would be welcome.

We have now included some views of the voxelized probability of occupancy for the granite intrusion and basement layers of runs B and D (new figure).

> P20.l24-25. You mention a number of interpolation techniques. Have you tried kriging, as it is widely used in geostatistics?

Our understanding is that kriging is synonymous with Gaussian process regression, so yes, in fact we use it here. We have inserted a reference to the term "kriging" in section 2.4 when we first mention the use of Gaussian processes for depth-to-boundary interpolation.

> P23.l16-17. I am not sure that I understand the meaning of this sentence. Please clarify. By gradients, do you refer to the jacobian matrix? Or am I missing something?

In this case we mean the derivatives of the prior and likelihood with respect to model parameters being sampled. We clarify in this section now that we mean derivatives of the posterior with respect to parameters rather than some spatial derivative, and mention Hamiltonian Monte Carlo (Duane et al. 1987; Neal 2011) and Riemannian manifold Monte Carlo (Girolami Calderhead 2011) by name as examples of proposals that need derivative information.

> P23.l30. the package proposed by de la Varga et al. (2018) offers the advantage of being open source but it is not the only one performing probabilistic geological modelling. For instance, other works using ideas introduced by Wellmann et al. (2010) such as Pakyuz-Charrier et al. (2018a) also achieves this.

We agree that both Pakyuz-Charrier papers are probabilistic, but as with other similar issues above, we also want to distinguish uncertainty propagation methodologically from sampling of the model posterior. MCUE is a fine solution if the only data are structural, but as mentioned in de la Varga Wellmann (2016), MCMC sampling of the posterior becomes necessary to fuse structural data with other types.

Besides the abovementioned paragraph in the Introduction describing MCUE as a branch of probabilistic modeling, we have updated the sentence mentioned here to give a more specific description of GemPy's advantages: "The GemPy package developed by de la Varga et al. (2018) makes an excellent start on a more general-purpose open-source code for 3-D geophysical inversions: it uses the implicit potential-field approach (Lajaunie 1997) to describe geological structures, includes forward-models for geophysical sensors, and is designed to produce posteriors that can easily be sampled by MCMC."

**Responses to Anonymous Referee 2**

> This study explores the influence of various practitioner decisions on MCMC posterior sampler efficiency for a geophysical joint inversion with a layered paramitrization; specifically, the influence several of proposal, prior, and likelihood function options. The tests are well designed and succeed in addressing the questions asked. I personally did not find much of the results and conclusions surprising, most of it could be deduced from purely theoretical grounds. However, the topic is important and this paper gives a good empirical basis from which future geophysical posterior sampling work can draw. On these grounds, I think it deserves to be published.

We thank the referee for their feedback and are pleased to hear that they recommend publication once their comments are addressed.

Most of the paper is well written and clear, with the introduction being the exception. It seems rushed and the odd use of Bayesian/statistical/probabilistic terminology (in the introduction only) suggests a lack of familiarity. I personally don't see the need for additional detail on the software use and implementation since that is not what the paper is about and if anything the scope should be more contained not expanded.

While a case could be made simply to cite the original software paper (McCalman et al 2014) regarding all such details, the fact that Referee 1 asked for more details suggest that our paper will be more accessible if at least an overview of the code and its performance is provided here.

The review of the MCMC literature review is extensive and was interesting to read. Complexity of the shape of the posterior is discussed several times but seldom in the context of previous work. For example, the non-linearity and complex correlations of physical parameters for non-unique magnetotelluric inversion is well known, but no overview is given here on that. I believe the discussion section could be improved by relating more to the known properties of the different geophysical forward problems.

I think the discussion section is needlessly bloated. Here the authors go into detail on many topics which the experiments shown here had no bearing on. Various things that could be done or might work are listed here which are in no way related to what the study presented actually did. I strongly suggest rewriting this section to be more on topic.

Our main aim in this work is to flag and address challenges for the uptake of Bayesian reasoning and MCMC in joint inversion methods for 3-D geological models, in ways we hope are accessible both to geoscientists and to statisticians. Some of the citations

suggested by Referee 2 involve some quite advanced methods, relating mostly to 1-D non-parametric inversions for single sensor types, and we are happy to acknowledge them. In contrast, Referee 1 focused on probabilistic methods for error propagation in geological models rather than on posterior sampling, and our impression is that posterior sampling for uncertainty quantification is still quite rare in this area because existing MCMC methods are still too costly. The forward model for each sensor contributes to the posterior shape, but so does the prior. Our discussion is a little more than a page long and addresses future directions. The work most directly corresponding to our direction is de la Varga Wellmann (2016) and de la Varga et al. (2018), whom we acknowledge and cite.

We recognize that this focus may not have come across well in the original Introduction, and believe that our revisions of the Introduction in response to Referee 1's comments make our intended contribution clearer.

> P1L12) What does "improve inversion results" refer to? Most readers would assume that it means a more accurate inversion. Since accuracy of results, compared to reality, is never quantified in this work, I don't see how this claim is backed up. One might argue that if true sensor noise levels are known, uninformative priors on them would only increase chances of their miss-estimation. Counterarguments based on model inadequacy could be raised of course, but these are not things that this study shed light on so please remove this claim.

Removed. If true sensor noise levels are known in detail then we agree that using an informative prior is more appropriate. Information about noise levels, however, is frequently not available in detail for public survey data our end users might want to fuse, nor was it available for the data set we used.

> P1L13-15) I do not see why this claim about using gradient information is

in the abstract. The statement is probably true, but this study did not show anything new to support it.

Removed.

> P2L25) It's not clear what is meant by posterior ensembles being a 'gold-standard'.

This is a normative statement from the statistical community. The true "gold standard" would be an analytic form for the exact posterior. MCMC, however, has theoretical guarantees to converge to the target distribution given enough computing time. We have removed the "gold-standard" wording and have updated the text as follows: "The output of a Bayesian method is also a probability distribution (the posterior) representing all values of system parameters consistent both with the available data and with prior beliefs. For complex statistical models the exact posterior cannot be expressed analytically; in such cases Monte Carlo algorithms, in particular Markov chain Monte Carlo (MCMC; Mosegaard 1995, Sambridge 2002) can provide samples drawn from the posterior for the purpose of computing averages over uncertain properties of the system."

> P2L29) Online updating is not necessary or sufficient for optimal for decision-making; these are separate things. The only relevance I can see here is that it could speed up decision making.

We agree this was poorly worded; we are referring to the potential for Bayesian updating and Bayesian optimization for acquisition of additional data. We have updated the text to read: "The inference also can be readily updated as new information becomes available, using the posterior for the previous inference as the prior for the next

one. This use of Bayesian updating allows automated decision-making about which additional data to take to minimize the cost of reducing uncertainty (Mockus 2013)."

> P3L3) What about overestimating uncertainties?

See our response to Referee 1 on a similar question. This sentence now reads: "Use of the inverse Fisher information matrix to describe posterior uncertainty implicitly assumes a single multivariate Gaussian mode; for posteriors with multiple modes or significant non-Gaussian tails, the inverse Fisher information provides a lower bound on the posterior variance (Cramer 1946; Rao 1945) and may be a significant underestimate."

> P3L4) The "no 'one-size-fits-all' solution exists" comment is very important. Perhaps give the reader some direction by citing something (e.g. Wolpert et al., 1997, No free lunch theorems for optimization: IEEE transactions on evolutionary computation, 1, 67-82.)

The sentence is referring to sampling, not optimization. There are several MCMC-related references that make this point and we have chosen a recent one: Green (2015) Bayesian computation: a summary of the current state, and samples backwards and forwards, Statistics and Computing July 2015, Volume 25, Issue 4, pp 835–862. We have also revised this sentence to further highlight the point: "Since the most appropriate sampling strategy may depend ont he characteristics of the posterior for specific problems, sampling methods must usually be tailored..."

> P4L21-29) This paragraph should probably lead with the last sentence (lines 27-29). The parts about deterministic inversion reads like an odd tangent and I didn't see the relevance and purpose of it until a second read through.

This is a good suggestion and we have revised the paragraph accordingly: "Although each of these elements has a correspondence to some similar model element in more traditional geophysical inversion literature (for example Menke et al 2018), interpreting model elements in terms of probability may motivate different mathematical choices from the usual non-probabilistic misfit or regularization terms." We also now mention the role of cross-validation in calibrating non-probabilistic regularization terms that do not arise in a Bayesian setting where a prior or hyperprior governs the extent of regularization, and add some citations.

> P7) I did not pick up on the fact that all your tests use PTMCMC until the second read-through; this section should probably make that more explicit.

Section 3 intro, paragraph 3 now starts: "All experiments use PTMCMC sampling, with 4 simultaneous temperature ladders... each with 8 temperatures, unless otherwise specified."

> P12L1-5) Where the seismic lines used to inform the layer interface Gaussian process variogram?

More information on the construction of the prior used to set up the original Moomba inversion is given in a NICTA technical report (Beardsmore, 2014), which we now cite at the beginning of section 2.5 in addition to the less complete conference papers. The original seismic lines are not explicitly mentioned in any of these sources, nor were we able to learn this from the original authors. We do now provide maps of the locations of sensor readings we used for the three sensors we actually include, in a new figure referenced at the beginning of section 3.

> P14L8) What is 'global posterior shape'? is it always defined?

We agree this wording is vague and have updated the sentence to be more specific: "The adaptive (anisotropic) Gaussian random walk (Haario et al. 2001), or aGRW, attempts to learn an appropriate covariance structure for a random walk proposal based on the past history of the chain."

P14) Could you have used the layer Gaussian process covariances directly to create a proposal function, it seems like that is what the CNp effectively does?

The proposal described in this comment is a sort of random walk with the same covariance as the prior. This is the behavior of pCN in the limit of small stepsize. In the limit of the medium-to-large stepsizes taken in the higher-temperature chains, pCN behaves very differently and samples more effectively than a random walk would, especially in high-dimensional spaces, a point we make when we introduce pCN in section 3.1 and made in more detail by Cotter et al. 2013.

> P16L4) "Figure 2 shows that iGRW and aGRW have more trouble travelling between different posterior modes than pCN" Tell us how you deduce this from that figure.

This is a fair point – the trace plots don't obviously support this conclusion. The worst-case autocorrelation and the differences in posterior weight between the two modes among repeat runs under similar conditions do support this conclusion, but we refer to these already in the preceding paragraphs. We have thus rephrased to say: "The different proposals vary in performance when hopping between modes despite the fact that all three proposals are embedded within a PTMCMC scheme..."

> P17) Please mention why MT noise levels were fixed.

We varied MT noise levels as well. We can see how this might not have been clear and so we now say: "the noise prior is set to $\alpha = 0.5$, $\beta = 0.05$ for all sensors (gravity, magnetic, and magnetotelluric)."

P21) Increasing the amount of data points by interpolation seems like a terrible idea. Why would anyone even attempt it? The observed effects should be obvious. If there are actual examples of Bayesian posterior analysis papers which do this, please cite one; otherwise, this seems like an odd and unnecessary test to include.

We agree that it's a terrible idea, and that nobody who is already thinking probabilistically about the data would be likely to do this. We were thinking specifically about the potential for uncritical use of re-gridded data by end users not already accustomed to Bayes or MCMC, especially when we (the statistician collaborators) learned that re-gridding was common for public geophysical survey data. In retrospect it would have been a fairer test to compare original to re-gridded measurements using actual widely adopted re-gridding methods, or to demonstrate new likelihoods for handling re-gridded potential field data.

This section isn't central to our results, though, and is more about pedagogy than development of new knowledge. At this stage we've decided to remove this section and replace it with a brief warning in section 2.1 ("The implicit assumption behind the use of mean square error...").

P21L16-19) I don't know about gravity and and magnetic, but Gaussian process likelihood functions have been used for MT and seismic MCMC. Relevant work should be cited here, E.g.:

Agostinetti, N. P., and A. Malinverno, 2010, Receiver function inversion by trans- dimensional Monte Carlo sampling: Geophysical Journal International, 181, 858–872. Bodin, T., M. Sambridge, H. TkalcĚĞic ÌĄ, P. Arroucau, K. Gallagher, and N. Rawlinson, 2012, Transdimensional inversion of receiver functions and surface wave dispersion: Journal of Geophysical Research: Solid Earth, 117. Xiang, E., R. Guo, S. E. Dosso, J. Liu, H.

Dong, and Z. Ren, 2018, Efficient hierarchical trans-dimensional Bayesian inversion of magnetotelluric data: Geophysical Journal International, 213, 1751–1767.

Also, there are ways to learn the correlation during sampling: Steininger, G., J. Dettmer, S. E. Dosso, and C. W. Holland, 2013, Trans-dimensional joint inversion of seabed scattering and reflection data: The Journal of the Acoustical Society of America, 133, 1347–1357.

We thank the referee for bringing these papers to our attention; we cite them now in the paragraph in section 2.1 regarding correlations in the likelihood.

P22L19) "The clearest lesson we can draw ..." I'm not sure why this is the lesson you lead with, in the introduction it was stated as known; almost every MCMC application to geophysics show this and it was not among the questions that your tests were set up to answer.

P22L20) "Our results were sensitive to ..." Each point raised in this sentence will be true for for any difficult posterior sampling problem. This is not a new result and this sentence adds nothing to the manuscript.

Since we view our contribution here as focused on the interaction between problem setup and sampling efficiency, we have replaced this text with a one-sentence introduction to the next paragraph: "Our experiments show concrete examples of how the efficiency of MCMC sampling changes with assumptions about the prior, likelihood, and proposal distributions for an Obsidian inversion, particularly as tight constraints on the solution are relaxed and uncertainty increases."

P22L34) I don't agree with the claim that either of these outcomes are counter-intuitive. Tighter constraints lead to narrower local optima, hence

more sampling is needed. Cauchy likelihood functions are more likely to give multi-modal posteriors than Gaussian likelihood functions, even for the most trivial problems (e.g. with just one parameter).

Removed the first part of this sentence re: whether intuitive or not, since this may depend on the reader; added mention of likelihoods ("but relaxing priors or likelihoods may sometimes widen...") to statement about relaxed constraints.

> P23L6-10) None of these three dot-point listed statements were informed by the experiments presented in this manuscript. The claims are also obvious and well known.

> P24L10) "However, proposals using gradients from auto-differentiation are probably needed to make further progress in this area." This claim, while probably true, is not really backed up by what is in the manuscript. Why is it listed as a conclusion?

Removed. This is really a statement about future work, which we leave in our Discussion section.

> P24L14) This is a trivial claim by itself. How can it help design future work. Will the better fit derived from uninformative priors lead to more accurate results in terms of uncertainty estimation. This ties in with my comment for P1L12.

We have updated the text to relate more specifically to a conclusion about what was done, again tying back to the sampling: "Hierarchical priors on observational noise provide a way to capture uncertainty about the weighting among datasets, although this may also make sampling more challenging as when priors on world parameters are relaxed."

[Figure]

P2417) Without some guiding principle for how to do the sub-sampling, this is not useful.

Removed (since the corresponding section has been removed).

TECHNICAL CORRECTIONS:

P2L20) "..., but about uncertainties." Awkward use of terminology, a Bayesian probability is an uncertainty and an assumption. Assumptions are specified as uncertainties quantified by probability distributions.

Changed to: "...not only about expected values or point estimates for system parameters, but about their beliefs regarding the true values of those parameters."

P2L26) "The posterior distribution is a representation of all possible outcomes and hence provides an internal estimate of uncertainty." The world parameterization is the representation of all 'possible' outcomes. What does the 'internal estimate' mean? This sentence is incoherent.

We have reworded this paragraph in addressing the "gold-standard" comment above.

P6L12) Spell out what SGR stands for here.

We define this acronym ("sequential geostatistical resampling") in the Introduction where it is first used.

P6L26-27) Grammar mistake.

Actually a LaTeX mistake; it seems GMD's LaTeX template doesn't support the form of the natbib command we used. We have changed to: "Parallel-tempered MCMC, or PTMCMC [ref], ..."

> P13L1) Table 1, what is N? First it was iteration count, then number of layers, then what? Readers shouldn't have to fish through the past 12 pages to find out.

We have relabeled this symbol "Nsamp" to distinguish it from other uses of N in the paper. We have also now included the symbols for each of these quantities in the table caption.

> P18L4) Grammar mistake.

Fixed; now reads: "may include systematic residuals..."